



# Directionally dependent Lambertian-equivalent reflectivity (DLER) of the Earth's surface measured by the GOME-2 satellite instruments

Lieuwe G. Tilstra, Olaf N. E. Tuinder, Ping Wang, and Piet Stammes

Royal Netherlands Meteorological Institute (KNMI), De Bilt, the Netherlands

**Correspondence:** Lieuwe G. Tilstra (tilstra@knmi.nl)

**Abstract.** In this paper we introduce the new concept of directionally dependent Lambertian-equivalent reflectivity (DLER) of the Earth's surface retrieved from satellite observations. This surface DLER describes Lambertian (isotropic) surface reflection which is extended with a dependence on the satellite viewing geometry. We apply this concept to data of the GOME-2 satellite instruments, to create a global database of the reflectivity of the Earth's surface, providing surface DLER for 26 wavelength

bands between 328 and 772 nm as a function of the satellite viewing angle via a second-degree polynomial parameterisation. The resolution of the database grid is 0.25 by 0.25 degrees but the real, intrinsic spatial resolution varies over the grid from 1.0 by 1.0 degrees to 0.5 by 0.5 degrees down to 0.25 by 0.25 degrees by applying dynamic gridding techniques. The database is based on more than ten years (2007–2018) of GOME-2 data from the MetOp-A and MetOp-B satellites.

The relation between surface DLER and surface BRDF is studied using radiative transfer simulations. For the shorter wave-

lengths ($\lambda < 500$ nm), there are significant differences between the two. For instance, at 463 nm the difference can go up to 6% at 30° solar zenith angle. The study also shows that, although BRDF and DLER are different properties, they are comparable for the longer wavelengths ($\lambda > 500$ nm). Based on this outcome, the GOME-2 surface DLER is compared with MODIS surface BRDF data from MODIS band 1 (centred around 645 nm), using both case studies and global comparisons. The conclusion of this validation is that the GOME-2 DLER compares well to MODIS BRDF, and that it does so much better than the

non-directional LER database. The DLER approach for describing surface reflectivity is therefore an important improvement over the standard isotropic (non-directional) LER approaches used in the past.

The GOME-2 surface DLER database can be used for the retrieval of atmospheric properties from GOME-2 and from previous satellite instruments like GOME and SCIAMACHY. It will also be used to support retrievals from the future Sentinel-5/UVNS satellite instrument.

## 1  Introduction

Most satellite retrievals of atmospheric composition require accurate information about the reflectivity of the Earth's surface to achieve accurate retrieval results. This includes the retrieval of trace gases such as ozone, $NO_2$, BrO, $CH_2O$, $H_2O$, $CO_2$, CO, and $CH_4$, and of cloud and aerosol information. To date, many of these retrievals use Lambertian surface reflection in the radiative transfer calculations and, consequently, adopt the use of Lambertian (isotropic) surface albedo climatologies.



Examples are the retrievals of $NO_2$ (e.g. Boersma et al., 2011; Bucsela et al., 2013), formaldehyde ($CH_2O$) (e.g. de Smedt et al., 2015; Hewson et al., 2015), and cloud products (e.g. Lelli et al., 2012; Veefkind et al., 2016). Although relying on Lambertian reflection is common practice, using a bi-directional reflectance distribution function (BRDF) (Nicodemus et al., 1992) to describe the surface reflectivity would be preferable. According to a recent study by Lorente et al. (2018), the simplification of

using Lambertian surface reflection can lead to errors of a factor of two in the surface reflection for vegetated surfaces.

Recently, several approaches have been introduced to address this issue. Examples of this are the introduction of geometry-dependent surface Lambertian-equivalent reflectivity (GLER) (Vasilkov et al., 2017; Qin et al., 2019) and similar work described in a recent paper by Loyola et al. (2020). In the GLER approach, surface BRDF information from the MODIS surface BRDF database (Gao et al., 2005) is used to calculate Lambertian surface albedo at 466 nm for land-covered satellite footprints

of the OMI instrument. The result is a Lambertian surface albedo, ready to be used in a radiative transfer code with Lambertian surface reflection, calculated for the exact scattering geometry of the OMI footprint. The advantage is that this Lambertian surface albedo is adjusted to the geometry of the observation, whereas the surface albedo available in the typical Lambertian surface albedo climatologies is more representative for the minimum value of the surface reflectivities that were observed (see e.g. Lorente et al., 2018; Liu et al., 2020) – and therefore underestimates the surface albedo for many of the scattering geome-

tries. The disadvantage of the GLER approach is that it, at least for land-covered scenes, depends fully on the MODIS surface BRDF database. This limits the spectral usage to the seven wavelength bands of the MODIS BRDF product. For the retrieval of $NO_2$ and of cloud properties from the $O_2$-$O_2$ band, both performed in the spectral regime close to 466 nm, this is not a problem – but for many other retrievals it is.

In this paper we introduce the directionally dependent Lambertian-equivalent reflectivity (DLER) of the Earth's surface

derived from GOME-2 observations. The surface DLER is retrieved as a function of the viewing geometry and therefore describes the anisotropy of the surface reflectivity. The DLER approach is very different than the GLER approach in that we perform a retrieval directly on (in this case GOME-2) instrument data, not relying on BRDF input (or any other input) from an external database. In this way the wavelength bands, 26 in total, can be chosen freely, allowing the resulting DLER database to support the retrieval of most atmospheric species. Another difference is that the directional dependence of the DLER is

provided as a parameterisation of the viewing angle. It is not mapped on a satellite footprint. The directional approach of the GOME-2 surface DLER is therefore applicable to all polar satellites with equator crossing times close to that of GOME-2 (09:30 LT). This includes satellite instruments like GOME and SCIAMACHY, GOME-2 itself, and the future Sentinel-5/UVNS instrument scheduled for launch in 2023.

Like the GLER, the DLER is a Lambertian property and therefore can be used in situations where radiative transfer cal-

culations include Lambertian surface reflection. The GOME-2 surface DLER database is an important improvement on the non-directional GOME-2 surface LER database that was described earlier (Tilstra et al., 2017). The transition from LER to DLER is the main topic of this paper including a study on the theoretical difference between DLER and BRDF. Other improvements to the database are also described in this paper.

The paper is structured in the following way. Section 2 introduces the theory behind Lambertian-equivalent reflectivity (LER)

and the new concept of directionally dependent LER (DLER). In Sect. 3 the theoretical difference between surface DLER and





surface BRDF is studied. Section 4 provides a short description of the GOME-2 instrument. The algorithm setup, atmospheric correction, and the theoretical background of the improved surface DLER retrieval algorithm are described extensively in Sect. 5. Section 6 presents results and provides examples of the anisotropy of the Earth's surface according to the new GOME-2 surface DLER database. In Sect. 7 the DLER database is compared to MODIS BRDF. Case studies and global comparisons

are both performed and the validation results are discussed. The paper ends with a summary and conclusions.

## 2   DLER theory

This section introduces the concept of a directionally dependent Lambertian-equivalent reflectivity (DLER) to describe the reflectivity of the Earth's surface. The following definition of the Earth reflectance is adopted in this paper:

$$R = \frac{\pi I}{\mu_0 E} \tag{1}$$

In Eq. (1), the symbol $I$ refers to the Earth radiance at the top-of-atmosphere (TOA), in $\mathrm{Wm^{-2}sr^{-1}nm^{-1}}$. The symbol $E$ refers to the incoming solar irradiance, perpendicular to the solar beam, at the TOA, and given in $\mathrm{Wm^{-2}nm^{-1}}$. The parameter $\mu_0$ is a shorthand for $\mu_0 = \cos\theta_0$, with $\theta_0$ the solar zenith angle. The shorthand for the viewing direction is $\mu = \cos\theta$, with $\theta$ the viewing zenith angle. The symbols for the viewing and solar azimuth angles are $\phi$ and $\phi_0$, respectively.

### 2.1   Lambertian-equivalent reflectivity

The focus of this paper is on Lambertian surface reflection in combination with clear-sky atmospheric conditions. For these conditions, there exists a simple relationship between the Earth reflectance $R$ and the (Lambertian) surface albedo $A_\mathrm{s}$ (Chandrasekhar, 1960):

$$R(\mu, \mu_0, \phi, \phi_0, A_\mathrm{s}) = R^0(\mu, \mu_0, \phi - \phi_0) + \frac{A_\mathrm{s} T(\mu, \mu_0)}{1 - A_\mathrm{s} s^\star} \tag{2}$$

In Eq. (2), the first term on the right is the so-called path reflectance $R^0$. This is the atmospheric contribution to the Earth

reflectance for a Rayleigh atmosphere which is bounded below by a non-reflecting surface. The second term in Eq. (2) is the surface contribution to the Earth reflectance. This term depends on the surface albedo $A_\mathrm{s}$, on the total transmission $T$ of the atmosphere, and on the spherical albedo $s^\star$. The property $s^\star$ is the spherical albedo of the Rayleigh atmosphere for illumination from below. The parameters $R^0$, $T$ and $s^\star$ can in principle be calculated using any radiative transfer model (see e.g. Tilstra et al., 2012).

From a given measured reflectance $R^\mathrm{obs}$, the surface albedo $A_\mathrm{s}$ can now be determined from Eq. (2):

$$A_\mathrm{s} = \frac{R_\lambda^\mathrm{obs} - R_\lambda^0}{T_\lambda(\mu, \mu_0) + s_\lambda^\star (R_\lambda^\mathrm{obs} - R_\lambda^0)} \tag{3}$$

When clear-sky conditions apply, the parameter $A_\mathrm{s}$ is the Lambertian-equivalent reflectivity (LER) of the surface.





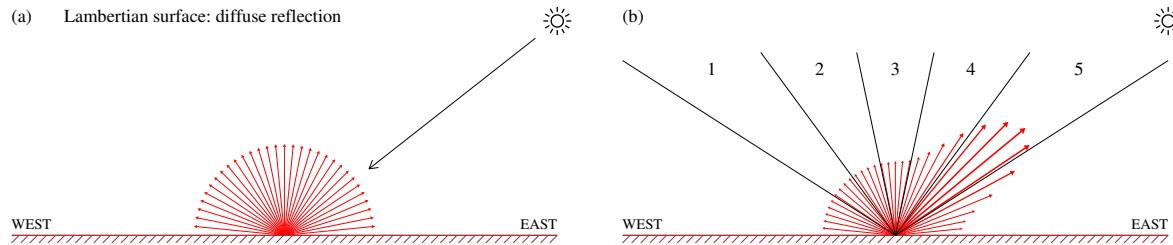

**Figure 1.** Left: Illustration of the principle of Lambertian (isotropic) surface reflection. Right: Surface reflection distribution with a retro-reflection lobe, representative for vegetation. In the DLER retrieval code, the orbit swath is divided into five viewing angle ranges and for each segment the surface LER is determined in the usual way.

## 2.2 Directionally dependent surface LER

Traditional, non-directional surface LER databases are built on the assumption that all surface types act as Lambertian reflectors. That is, one assumes that the amount of light being reflected by the surface does not depend on the direction of incoming and reflected light. This is illustrated in Fig. 1a. The Lambertian assumption is, unfortunately, in many cases not justified.

A more realistic description of the reflective properties of a surface requires a bi-directional reflectance distribution function (BRDF) (Nicodemus et al., 1992). The BRDF is a function of the incoming and outgoing directions. A hypothetical surface BRDF is shown in Fig. 1b. Here, the surface BRDF contains a retroreflection lobe, resulting from increased reflection by vegetation in the backscattering direction.

In the retrieval algorithm of the traditional GOME-2 surface LER database (Tilstra et al., 2017), grid cells acts as storage

containers in which all observations with fitting geolocation are stored, irrespective of viewing geometry and scene conditions. Statistical methods are then employed to identify the cloud-free scenes. For the *directional* GOME-2 surface DLER, the grid cell container is split into five sub-containers, each representing a certain viewing angle range (see Fig. 1b). The traditional retrieval algorithm is then run five times, deriving surface LER for each of the five viewing angle containers. The viewing angle dependence can then be analysed. This procedure is explained in Fig. 2.

The coloured circles in Fig. 2 represent the surface LER retrieved by GOME-2 for a grid cell over the Sahara desert, for the five viewing angle containers and for the 26 wavelength bands defined in Sect. 5.1. The viewing angle $\theta_v$ presented on the horizontal axis is defined as:

$$\theta_v = \begin{cases} -\theta & , \quad \text{for the east viewing direction} \\ \theta & , \quad \text{for the west viewing direction} \end{cases} \tag{4}$$

The centres of the viewing angle containers are indicated by the dotted vertical lines. Parabolic curves are fitted to the five

retrieved surface LER values for all wavelength bands. Labels are provided for most of the wavelength bands.

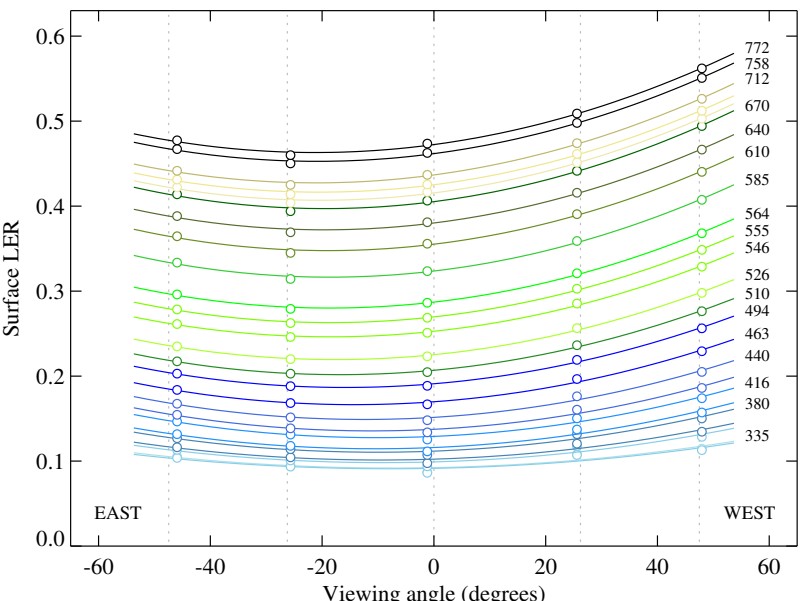

**Figure 2.** Surface LER retrieved for a grid cell over the Sahara desert for the five viewing angle containers (indicated by the circles). The associated viewing angle $\theta_v$ is plotted on the horizontal axis. The vertical dotted lines indicate the centres of the viewing angle ranges. Colours indicate the selected wavelength bands. The curves are parabolic fits through the data points.

From Fig. 2 it can be concluded that there is a clear dependence on $\theta_v$. The parabolic fits suggest that the dependence may be parameterised as a function of the viewing angle $\theta_v$ in the following way:

$$A_{\mathrm{DLER}} = A_{\mathrm{LER}} + c_0 + c_1 \cdot \theta_v + c_2 \cdot \theta_v^2 , \tag{5}$$

where $\theta_v$ is negative on the east side of the orbit swath, and positive on the west side of the orbit swath, see Eq. (4). The coefficients $c_0$, $c_1$, and $c_2$ are wavelength dependent and are calculated for each grid cell, provided that all five viewing angle segments are sufficiently filled with observations. For water bodies the coefficients $c_0$, $c_1$, and $c_2$ are set to zero, because surface DLER and BRDF cannot be cast into a climatology easily for water surfaces. This is because of the strong dependence on the viewing and solar angles for sun glint conditions and because of the dependence on parameters such as wind speed and chlorophyll concentration. The provided surface albedo over water is therefore the standard minimum LER and more representative for the diffuse component of the surface reflection (Liu et al., 2020).

## 3  Theoretical study: DLER versus BRDF

In this section we study the theoretical difference between surface DLER and surface BRDF. As explained in Sect. 1, BRDF and DLER are fundamentally different properties and as such cannot be expected to yield the same values or to take over the role of the other in radiative transfer calculations. Nevertheless, as the results in this section will show, for certain wavelength regimes



BRDF and DLER are numerically comparable. This allows practical applications and validation of DLER by comparison with BRDF (and vice versa).

## 3.1 MODIS BRDF

The MODIS Ross–Li surface BRDF model is a linear kernel-based BRDF model used to describe the surface reflectance of

land surfaces. The surface anisotropy is described by two geometry-dependent kernels, which have to be combined with the provided kernel coefficients if one wants to calculate the BRDF. The Li–Sparse kernel $K_{\mathrm{geo}}$ is the geometric kernel, which describes the contribution of sunlit and shaded parts of a scene due to the presence of three-dimensional object, typically trees. The Ross–Thick kernel $K_{\mathrm{vol}}$ is the volumetric kernel, which describes the smaller-scale variation of the leaf canopy, i.e., the orientation of the leafs themselves.

The geometric and volumetric kernels are independent on wavelength. The wavelength dependence of the BRDF is contained entirely in the kernel coefficients. The expression for the surface reflectivity is (Strahler et al., 1999):

$$
\begin{aligned}
A_{\mathrm{g}}(\lambda, \theta, \theta_0, \phi - \phi_0) = {} & f_{\mathrm{iso}}(\lambda) \\
& + f_{\mathrm{vol}}(\lambda) \cdot K_{\mathrm{vol}}(\theta, \theta_0, \phi - \phi_0) \\
& + f_{\mathrm{geo}}(\lambda) \cdot K_{\mathrm{geo}}(\theta, \theta_0, \phi - \phi_0)
\end{aligned}
\tag{6}
$$

The exact expressions needed to calculate $K_{\mathrm{vol}}$ and $K_{\mathrm{geo}}$ are provided in Appendix A. The coefficients $f_{\mathrm{iso}}$, $f_{\mathrm{vol}}$, $f_{\mathrm{geo}}$ are the kernel coefficients of the isotropic, volumetric, and geometric contributions.

## 3.2 DLER and BRDF model calculations

For our model calculations we make use of the DAK radiative transfer code which will be described extensively in Sect. 5.3. Here we make use of surface reflection defined by a BRDF instead of Lambertian surface reflection as described by Lorente et al. (2018). We thereto provide DAK the three kernel coefficients ($f_{\mathrm{iso}}$, $f_{\mathrm{vol}}$, $f_{\mathrm{geo}}$) as defined in the MODIS ATBD (Strahler et al., 1999). Using this setup, the TOA reflectance is calculated at a number of wavelengths, for the VZA and SZA nodes $\theta$

and $\theta_0$ that are also part of the LUTs described in Sect. 5.3, and for 360 equidistant values of the relative azimuth angle $\phi - \phi_0$. The surface elevation is set to zero (sea level) and the ozone column to 350 DU. As before, cloud and aerosols are not included. The calculations are performed monochromatically.

    Next, the surface DLER is retrieved from the simulated TOA reflectances using a similar setup as the one described in Sect. 2. The only difference here is that the input reflectances are not measured but simulated by DAK, i.e., they are based on

surface reflection described by the BRDF kernel coefficients ($f_{\mathrm{iso}}$, $f_{\mathrm{vol}}$, $f_{\mathrm{geo}}$). The differences between BRDF and DLER for all angles $\theta$, $\theta_0$, and $\phi - \phi_0$ are then analysed as a function of wavelength.

## 3.3 Analysis and discussion

The results for 772 nm are presented in Fig. 3 in the form of polar plots. The solar zenith angle was set to $32°$. The left upper window presents the BRDF, characterised by the kernel coefficients ($f_{\mathrm{iso}}$, $f_{\mathrm{vol}}$, $f_{\mathrm{geo}}$) = (0.36, 0.24, 0.03). These kernel





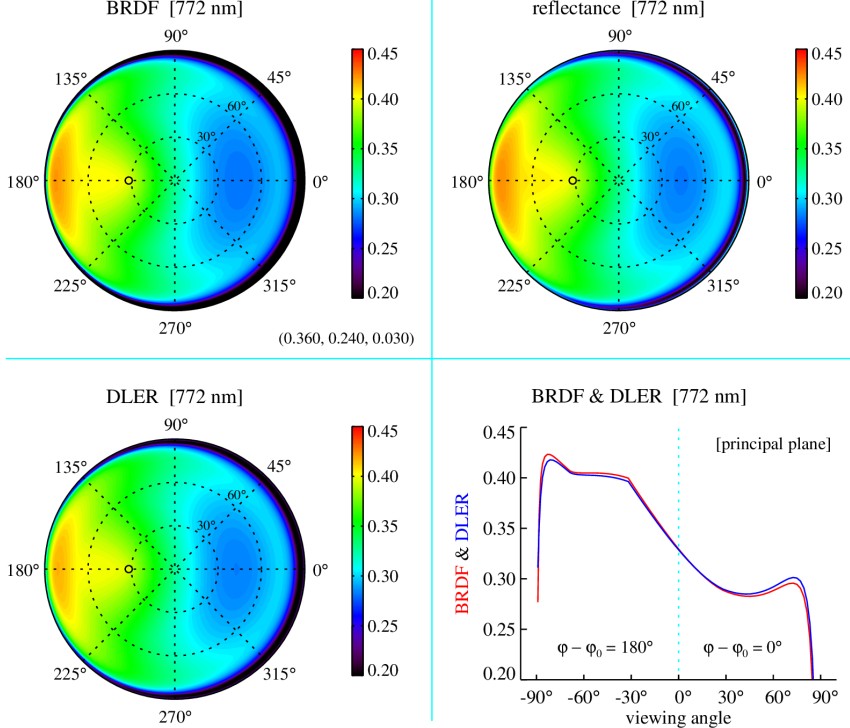

**Figure 3.** Top row: Surface BRDF at 772 nm for a solar zenith angle of $32°$, and the resulting simulated TOA reflectance. The BRDF kernel coefficients ($f_{\text{iso}}$, $f_{\text{vol}}$, $f_{\text{geo}}$) were set to (0.36, 0.24, 0.03), representative for vegetation. Bottom row: Retrieved surface DLER and BRDF/DLER in the principal plane. In the principal plane, where $\phi - \phi_0$ is $0°$ or $180°$, exact backscattering occurs at a viewing angle of $-32°$.

coefficients are representative for vegetated surfaces such as forests (Lorente et al., 2018). The right upper window shows the TOA reflectance calculated by the DAK RTM. Note that the reflectance is similar to the BRDF. The left bottom window presents the retrieved DLER. The differences between BRDF and DLER appear to be small. This is confirmed by the bottom right window, which presents the BRDF (green curve) and DLER (blue dotted curve) inside the principal plane ($\phi - \phi_0 = 0°$

5  or $180°$). Differences are found, but they are small even for large viewing zenith angles.

These results are not unexpected, because at 772 nm the Rayleigh optical thickness is quite low (about 0.02), so scattering in the atmosphere is relatively weak. This means that (single) surface reflection dominates and that the reflectance at the TOA is similar to the BRDF of the surface. Moreover, in this situation the retrieved surface DLER and BRDF are similar. Note that the behaviour of the BRDF for extreme viewing zenith angles in the forward scattering direction is suspicious, because the

10  BRDF becomes negative for viewing zenith angles close to $90°$. In the DAK RTM, the surface BRDF is therefore not allowed to become negative.





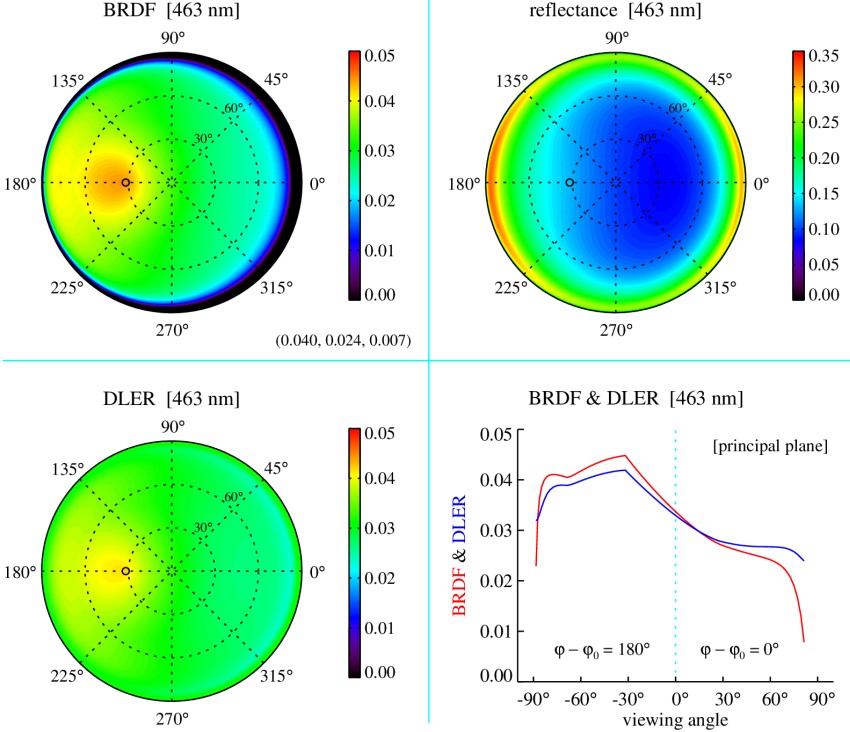

**Figure 4.** Similar to Figure 3 but now for 463 nm. Note that the BRDF kernel coefficients are different from the ones in Figure 3. The relative differences between DLER and BRDF are now larger, especially near the "hot spot" and for large viewing zenith angles.

The situation changes quite a bit at 463 nm (see Fig. 4). Figure 4 shows that at this wavelength the typical value of the BRDF is much lower than at 772 nm. Because of increased Rayleigh scattering in the atmosphere, the TOA reflectance is now very different from the surface BRDF. The retrieved DLER shows quite some differences compared to the BRDF. This is caused by an increased occurrence of scattering in the atmosphere (Rayleigh optical thickness $\sim$0.2 at 463 nm) and multiple scattering

via the surface. The effects may seem to be modest in an absolute sense, but relative to the typical value of the BRDF ($\sim$0.03) they can be quite large, depending on the viewing and solar angles that are involved. For instance, inside the "hot spot" the differences are 0.003, where the BRDF is 0.045 and the DLER is 0.042. The differences therefore can go up to 6% at this wavelength.

Based on the results we can distinguish three wavelengths regimes. For $\lambda > 1000$ nm the functional behaviour of DLER and

10 BRDF is nearly identical and interchangeable. For $500 < \lambda < 1000$ nm the DLER and BRDF values are similar, and DLER and BRDF can be interchanged in most practical situations. For example, at 555 nm the difference in the hot spot region is less than 3% for $\theta_0 = 32°$ and less than 8% for $\theta_0 = 60°$. For $\lambda < 500$ nm, however, DLER and BRDF differ by too much for them to take over each other's role. For example, in the UV at 380 nm the differences go up to 12% for $\theta_0 = 32°$ and even 30% for $\theta_0 = 60°$. It should be noted that the absolute differences in these cases are small (maximum on the order of 0.01). Please note



that vegetated surfaces have a relatively strong surface anisotropy compared to most other surface types such as desert. The provided numbers therefore represent worst-case situations.

Depending on the application, using BRDF instead of DLER, or DLER instead of BRDF, can be acceptable even below 500 nm. For the validation study presented in Sect. 7 we will, however, restrict ourselves to $\lambda > 500$ nm.

## 4 Description of GOME-2

GOME-2 (Munro et al., 2016) is the successor of the Global Ozone Monitoring Experiment (GOME) (Burrows et al., 1999). It is a remote sensing spectrometer that measures the Earth's radiance and the solar irradiance, covering the wavelength range between 240 and 800 nm. The spectral resolution varies between 0.3 and 0.5 nm. Like its predecessor GOME, GOME-2 performs scans of the Earth in a motion from east to west in 4.5 s (forward scan) and back from west to east in 1.5 s (backscan).

This motion is achieved via the rotation of an internal scanner mirror. The orbit swath is 1920 km wide and the measurement footprint in the forward scan is 80 km × 40 km (across track × along track). In about 1.5 days every location on the Earth's surface is observed.

Next to the spectral measurements of radiance and solar irradiance, also the polarisation of the light is measured. This is done by on-board polarisation measurement devices (PMDs) which measure the state of atmospheric polarisation in 15 wavelength

bands. The polarisation information is subsequently used to perform a correction for polarisation on the detected signals. Note that GOME-2 is sensitive to the polarisation of the incoming light since it is not equipped with a polarisation scrambler. The information from the PMD bands was used by us to also derive a surface DLER database based on the PMD bands. This PMD-based database is not described explicitly in this paper.

The first GOME-2 instrument was launched on 19 October 2006 as part of the MetOp-A satellite platform. Identical ver-

sions of the first GOME-2 instrument were launched on board the MetOp-B and MetOp-C satellites, with launch dates of 17 September 2012 and 7 November 2018, respectively. All three MetOp satellites were put into near-polar, Sun-synchronous orbits at an altitude of 820 km and with an orbital period of about 101 minutes. The local equator crossing time is 09:30 LT for the descending node, for all three satellite platforms but with different phasing. The MetOp series of satellites is expected to continue operations beyond the year 2027.[1]

The GOME-2 instruments were designed to perform global observations of trace gases for environmental and meteorological applications and climate monitoring. Trace gases that are retrieved are ozone, $NO_2$, $BrO$, $SO_2$, $HCHO$, $OClO$, and $H_2O$. Next to trace gases also cloud, aerosol, and surface properties are retrieved. A complete overview of the available GOME-2 products is presented in Hassinen et al. (2016).

---

[1]The orbit of MetOp-A is drifting since June 2017 and the satellite will be decommissioned in November 2021.

## 5  Algorithm setup and atmospheric correction

The DLER algorithm setup is in many aspects similar to the LER algorithm setup described in Tilstra et al. (2017). That is, the Earth reflectance spectrum is transformed into a number of reflectance bands, which are converted into scene LER values by applying the atmospheric correction outlined in Sect. 2.1. After these steps, the observed scene LER values of a specific

month (but from all available years) are distributed onto a latitude/longitude grid which represents the Earth's surface for that specific calendar month. For each grid cell the distribution of scene LER values is analysed in two ways. In the first method the so-called MIN-LER is retrieved, which is the 1% cumulative value of the scene LER distribution. The second method retrieves the so-called MODE-LER field. The MODE-LER is found from the mode of the scene LER distribution, which is a well-defined maximum for arid (desert) surfaces and snow/ice surfaces. For the other surfaces types, the mode cannot be

used and the MIN-LER result is copied. Both MIN-LER and MODE-LER fields are present in the surface LER database. The above steps and procedures have been described extensively in Tilstra et al. (2017). There are, however, a number of important improvements and extensions in the current algorithm:

1. The list of wavelength bands was extended with wavelength bands at 328, 585, 685, 697, and 712 nm. The wavelength bands at 685, 697, and 712 nm were introduced specifically to support cloud and aerosol retrieval near the $O_2$-B band,

as explained in Desmons et al. (2019). See Sect. 5.1 and Table 1 for details.

2. Spectral calculations were introduced for some of the wavelength bands and absorption by oxygen and water vapour was included, in the way described in Sect. 5.2.

3. The spatial resolution of the database fields was increased using dynamic gridding. This dynamic gridding approach is explained in Appendix B.

4. Data from both MetOp-A and MetOp-B were used, from the period 2007–2018, covering more than 10 years of observations. The larger amount of data used is beneficial for the quality of the climatology.

5. The database now offers directionally dependent surface LER (DLER). This means that the anisotropy of the surface reflection, often called the BRDF effect, is contained in (and described by) the DLER database.

### 5.1  Selection of wavelength bands

Table 1 provides a list of the chosen wavelength bands as well as their central wavelength and bandwidth. Note that the wavelength bands at 328, 585, 685, 697, and 712 nm were not present in the previous version of the GOME-2 surface LER database (Tilstra et al., 2017, Table 2). Most of the wavelength bands are one nm wide. The wavelength bands are therefore narrow enough to be considered monochromatic, but also wide enough to effectively minimise the impact of the Ring effect (Chance and Spurr, 1997). The reflectances for the wavelength bands are calculated from the reflectances measured by the

individual detector pixels that fall within the wavelength band. A boxcar weighting function $w$ is applied to each detector pixel





**Table 1.** Definition of the wavelength bands and details of the radiative transfer calculations for atmospheric correction.

| Wavelength band | 328 | 335 | 340 | 354 | 367 | 380 | 388 | 416 | 425 | 440 | 463 | 494 | 510 |
|---|---|---|---|---|---|---|---|---|---|---|---|---|---|
| Instrument channel | 2 | 2 | 2 | 2 | 2 | 2 | 2 | 3 | 3 | 3 | 3 | 3 | 3 |
| Central wavelength (nm) | 328.0 | 335.0 | 340.0 | 354.0 | 367.0 | 380.0 | 388.0 | 416.0 | 425.0 | 440.0 | 463.0 | 494.0 | 510.0 |
| Bandwidth (nm) | 1.0 | 1.0 | 1.0 | 1.0 | 1.0 | 1.0 | 1.0 | 1.0 | 1.0 | 1.0 | 1.0 | 1.0 | 1.0 |
| Spectral/monochromatic | S | M | M | M | M | M | M | M | M | M | M | M | M |
| Ozone absorption | + | + | + | + | + | + | + | + | + | + | + | + | + |
| $NO_2$ absorption | + | + | + | + | + | + | + | + | + | + | + | + | + |
| $O_2$-$O_2$ absorption | + | + | + | + | + | + | + | + | + | + | + | + | + |
| Oxygen absorption | – | – | – | – | – | – | – | – | – | – | – | – | – |
| Water vapour absorption | – | – | – | – | – | – | – | – | – | – | – | – | – |
| Wavelength band | 526 | 546 | 555 | 564 | 585 | 610 | 640 | 670 | 685 | 697 | 712 | 758 | 772 |
| Instrument channel | 3 | 3 | 3 | 3 | 3 | 4 | 4 | 4 | 4 | 4 | 4 | 4 | 4 |
| Central wavelength (nm) | 526.0 | 546.0 | 555.0 | 564.0 | 585.0 | 610.0 | 640.0 | 670.0 | 685.0 | 696.9 | 712.0 | 758.0 | 772.0 |
| Bandwidth (nm) | 1.0 | 1.0 | 1.0 | 1.0 | 1.0 | 1.0 | 1.0 | 1.0 | 1.0 | 0.2 | 1.0 | 1.0 | 1.0 |
| Spectral/monochromatic | M | M | M | M | M | M | M | M | M | S | S | S | S |
| Ozone absorption | + | + | + | + | + | + | + | + | + | + | + | + | + |
| $NO_2$ absorption | + | + | + | + | + | + | + | + | + | + | + | + | + |
| $O_2$-$O_2$ absorption | + | + | + | + | + | + | + | + | + | + | + | + | + |
| Oxygen absorption | – | – | – | – | – | – | – | – | – | + | + | + | + |
| Water vapour absorption | – | – | – | – | – | – | – | – | – | + | + | – | – |

The reflectance calculations are performed using spectral band integration or monochromatically. For all wavelength bands absorption by ozone, $NO_2$, and $O_2$-$O_2$ is included. Absorption by oxygen and/or water vapour is included for only some of the wavelength bands.

reflectance. This weighting function is defined as:

$$w_i^j = \begin{cases} 1 & , \quad \text{for } |\lambda_i - \lambda_j^c| \le \omega_j \\ 0 & , \quad \text{for } |\lambda_i - \lambda_j^c| > \omega_j \end{cases} \qquad (7)$$

In this equation, $\lambda_i$ is the wavelength of detector pixel $i$, $\lambda_j^c$ is the central wavelength of wavelength band $j$, and $2\omega_j$ is the width of wavelength band $j$. Normalisation of the resulting band reflectance is performed by dividing the result with the number of participating detector pixels, denoted by $N_j$.

Most of the wavelength bands are positioned in the continuum parts of the spectrum, avoiding absorption bands as much as possible. This is essential, because having to take absorption by atmospheric species into account complicates the radiative transfer calculations considerably. For wavelength bands located in the continuum monochromatic simulations are sufficient. This is not the case for a number of wavelengths bands which are affected too much from absorption by trace gases. These





wavelength bands (see Table 1) require a spectral handling of the radiative transfer calculations. This approach is described in Sect. 5.2.

## 5.2 Absorption by trace gases

Three examples of the impact of absorption by oxygen, water vapour and ozone are given in Fig. 5. The top panel shows the
situation for the wavelength band at 758 nm, positioned just in front of the $O_2$-A band, while the middle panel shows the situation for the wavelength band near 697 nm, which is spectrally surrounded by water vapour absorption lines. The bottom panel presents the situation for the wavelength band at 328 nm, where ozone absorption is quite variable over the extent of the wavelength band. The black curves represent the simulated reflectance spectra. These spectra were calculated for clear-sky conditions, for a surface albedo $A_s = 0.5$ at sea level, for nadir view and local noon ($\theta = \theta_0 = 0°$), for an ozone column of
350 DU, and for a water vapour column of $4.0 \, \mathrm{g \, cm^{-2}}$. For comparison, the horizontal green curves represent the reflectance spectra without taking absorption by oxygen and/or water vapour into account in the radiative transfer calculations.

For the wavelength band at 758 nm the impact of oxygen absorption is obviously very small. On the other hand, for the wavelength band near 697 nm the impact of water vapour absorption is much larger. A monochromatic calculation is clearly not sufficient in this case. To proceed, we first define $G_j(\lambda)$, the spectral response function of wavelength band $j$, as a weighted
superposition of the slit functions of the individual detector pixels by using the boxcar weighting defined in Eq. (7). That is, for the response function $G_j$ we have

$$G_j(\lambda) = \frac{1}{N_j} \sum_{i=1} w_i^j \cdot S_i(\lambda) \,, \tag{8}$$

where $S_i$ is the normalised slit function of detector pixel $i$ from the appropriate spectral band and $N_j$ the number of detector pixels that make up wavelength band $j$. The resulting response functions $G_{758}$, $G_{697}$, and $G_{328}$ are presented in Fig. 5 as
blue curves, in arbitrary units. The vertical green lines indicate the wavelengths $\lambda_i$ of the detector pixels that contribute to the reflectance of the wavelength band. Next, we calculate the simulated band reflectance. For this we first need to simulate the spectrum surrounding the wavelength band at a high spectral resolution. We use a spectral resolution of 0.01 nm. The spectral sampling is then increased by a factor of 100 using Akima interpolation (Akima, 1970). This allows for accurate numerical integration and the resulting expression for the simulated band reflectance is

$$R_j^{\mathrm{sim}} = \sum_k \Delta\lambda \cdot G_j(\lambda_k) \cdot R^{\mathrm{sim}}(\lambda_k) \,, \tag{9}$$

where the summation over $k$ involves a summation over the wavelengths $\lambda_k$ and $\Delta\lambda = 10^{-4} \, \mathrm{nm}$.

The impact of neglecting absorption by oxygen and/or water vapour and using monochromatic calculations can now be calculated. For the 758 nm case given in Fig. 5 this effect is only 0.003 on the reflectance and about the same for the surface LER. This wavelength band could therefore be treated monochromatically. For the 697 nm case, however, the effect is 0.018,
which is too high to justify monochromatic calculations. For the 328 nm wavelength band, adopting monochromatic calculation would lead to an error of −0.027. For the wavelength bands at 328, 697, 712, 758, and 772 nm we use Eq. (9) to calculate the



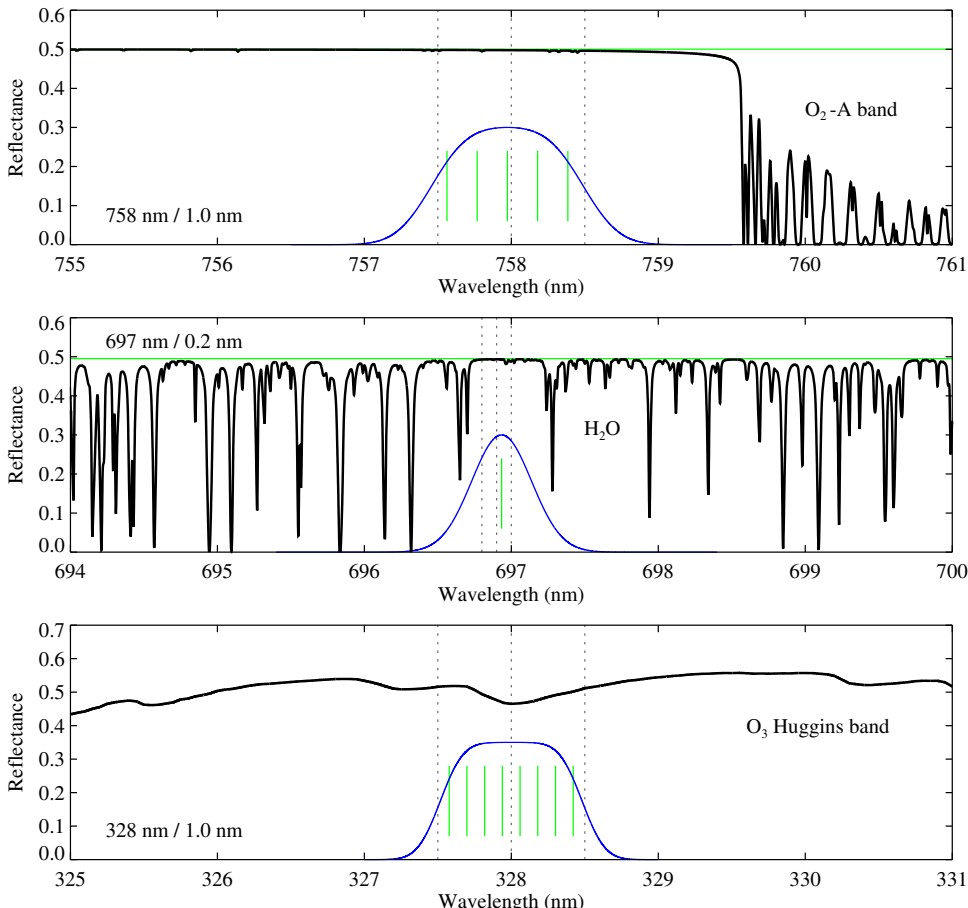

**Figure 5.** Simulated reflectance spectra (in black) relevant to the wavelength bands at 758, 697, and 328 nm. The upper panel shows the very small impact of oxygen absorption near 758 nm, while the middle panel shows the larger impact of water vapour absorption around 697 nm. The bottom panel presents the situation of ozone absorption near 328 nm. For comparison, the horizontal green curves indicate reflectance spectra simulated without absorption by oxygen and water vapour. The vertical green lines indicate the spectral positions of the detector pixels that make up the wavelength bands (see Table 1). The blue curves indicate the response functions of the wavelength bands, based on the indicated detector pixels and their individual slit functions.

simulated band reflectance. In Table 1 this is indicated by the label "S" in the fifth row. For the other wavelength bands we use monochromatic calculations, indicated by "M" in the fifth row of Table 1.

## 5.3 Radiative transfer calculations and LUTs

For the radiative transfer calculations we make use of the radiative transfer code "Doubling-Adding KNMI" (DAK) (de Haan, 1987; Stammes, 2001). The DAK code is able to calculate all four components of the Stokes vector (van de Hulst, 1981;





Hovenier et al., 2004) and in its minimal setup it features molecular scattering and Lambertian surface reflection, but the user can decide to include many other features such as scattering by clouds and/or aerosols, absorption by various trace gases, and surface reflection defined by a BRDF. The extension to BRDF was described by Lorente et al. (2018). In the calculations we did not include clouds and aerosols and adopted Lambertian surface reflection. Polarisation is included in the calculations. We used a standard Mid-Latitude Summer (MLS) atmosphere (Anderson et al., 1986) for the atmospheric profiles, and included absorption by ozone, $NO_2$, and $O_2$-$O_2$ for all wavelength bands. For some of the wavelength bands we also included absorption by oxygen and water vapour (see Table 1).

For all 26 wavelength bands look-up tables (LUTs) were created. The LUTs were made for 7 ozone column values (50, 200, 300, 350, 400, 500, and 650 DU), for 10 surface heights (ranging from 0 to 9 km in steps of 1 km), for water vapour columns of 0 and $4\,\mathrm{g\,cm^{-2}}$, and for 42 non-equidistant values of $\mu$ and $\mu_0$. The dependence on the relative azimuth angle $\phi - \phi_0$ can be treated analytically. To explain, because the simulations represent clear-sky Rayleigh atmospheres, the Fourier expansion of the reflectance in terms of the relative azimuth angle $\phi - \phi_0$ ends after only three terms. More specifically, we have

$$R^0 = a_0(\mu, \mu_0) + \sum_{i=1}^{2} 2a_i(\mu, \mu_0) \cos i(\phi - \phi_0) \,. \tag{10}$$

We therefore do not store the reflectances $R^0$ in the LUTs but instead store the Fourier coefficients $a_0$, $a_1$, and $a_2$. The reflectance $R^0$ can be calculated from these. The LUTs contain the parameters $a_0$, $a_1$, $a_2$, $T$, and $s^\star$.

## 6 Results

### 6.1 Surface anisotropy

Examples of the magnitude of the DLER surface anisotropy in GOME-2 data are provided in Fig. 6. The parameter plotted is the surface anisotropy parameter, defined here as the difference between the GOME-2 surface DLER at viewing angles $\theta_v$ of $+45°$ (west viewing direction) and $-45°$ (east viewing direction). For the GOME-2 orbit this parameter is a good indicator for the magnitude and range of the surface anisotropy in the GOME-2 orbit swath. The results in Fig. 6 are presented for calendar month March and for the wavelength bands at 772, 670, and 555 nm. At 772 nm the surface anisotropy parameter can be as large as 0.2 for vegetated areas. For the typical desert areas the differences are much smaller (0.05–0.10) because non-vegetated surfaces are usually more isotropic than vegetated areas. The surface anisotropy parameter for snow/ice surfaces has the opposite sign. The values over the vegetated areas correspond to percentages of 50–125%, in agreement with what was found already by Lorente et al. (2018). The magnitude of the surface anisotropy which is present in the GOME-2 surface DLER is therefore quite substantial.

At 670 and 555 nm the surface anisotropy parameter over vegetation is much lower than at 772 nm. However, this is mainly caused by the fact that the surface albedo at these wavelength bands is also much lower than at 772 nm. The percentages are more or less the same, in the range of 50–125%. For desert areas the anisotropy parameter is slightly smaller at 670 and 555 nm than at 772 nm. However, the percentages are similar for all three wavelength bands, about 10–20%. For snow/ice surfaces the

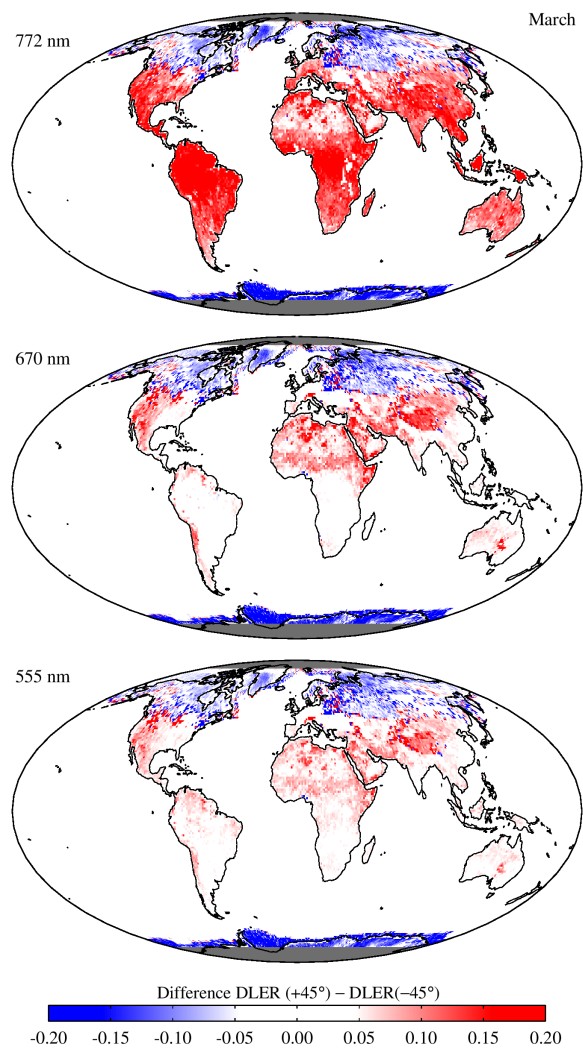

**Figure 6.** Global maps of the surface anisotropy parameter, defined as the difference between GOME-2 surface DLER at viewing angles of $+45°$ and $-45°$, for calendar month March and for 772, 670, and 555 nm. The surface anisotropy can be large, especially for vegetated surfaces.

anisotropy parameter is not depending much on the wavelength. For 555 nm the values are slightly smaller. This is probably because of increased Rayleigh scattering, which results in a more diffuse illumination of the surface. The surface anisotropy parameter varies mostly between $-0.1$ and $-0.3$ for snow/ice surfaces.





**Table 2.** Definition of the surface type regions studied in Fig. 7. The symbol "–" indicates that no constraint was set on the longitude.

| Description | Matthews land type | Latitude range | Longitude range |
| --- | --- | --- | --- |
| Sahara desert | 30 | 16–27°N | 12°W–15°E |
| Arabian Peninsula | 30 | 15–34°N | 37–61°E |
| Australian desert | 30 | 15–30°S | 114–145°E |
| Antarctica | 31 | 73–85°S | 0–45°E |
| Greenland | 31 | 70–80°N | 31–48°W |
| Amazonian tropical rainforests | 1 | 15°S–10°N | 40–85°W |
| Asian (sub-)tropical forests | 2,5,7,9 | 10–35°N | 70–125°E |
| Deciduous forests | 9–11 | 0–40°N | – |
| Grasslands | 23–28 | 35°S–35°N | – |

## 6.2 Dependence on surface type and time

The directional dependence of the surface DLER was studied for the nine surface types defined in Table 2. All nine surface types represent land surfaces. Constraints were set on latitude and longitude and, more importantly, on surface type using the Matthews land usage database (Matthews, 1983). Furthermore, coastal areas were excluded and so were all grid cells that

contained snow/ice, except for the "Antarctica" and "Greenland" surface types. The results are summarised in Fig. 7, which presents the GOME-2 surface DLER as a function of the viewing angle $\theta_v$ defined in Eq. (4). The solid coloured curves represent the surface DLER at 772 nm, averaged over the grid cells of the surface type region as defined in Table 2. The legend in the "Antarctica" window explains to which months the curves belong. The grey curves in Fig. 7 are there to provide an indication of the spread in the surface DLER. The spread is defined as 2.35 times the standard deviation in the data. The legend

in the "Greenland" window explains to which months the grey curves belong.

      The three desert surface types ("Sahara desert", "Arabian Peninsula", and "Australian desert") show a very similar dependence on the viewing angle. The overall dependence agrees well with that of the single grid cell shown in Fig. 2. The Australian desert deviates slightly from the other two desert regions because of the much lower values and the larger variability w.r.t. the calendar month. The latter observation may be partly explained by the different solar zenith angles but more likely it is caused

by the fact that the Australian desert contains more vegetation than the other two desert surface types. The snow/ice surface types ("Antarctica" and "Greenland") show that the highest value for the surface DLER is reached for the east-looking direction, and not for the west-looking direction as for the other surface types presented in Fig. 7. There is, at least for Antarctica, a mild dependence on the calendar month. Note that for certain months the surface DLER results are not plotted for these regions. This is because the regions are covered in polar night during these months. The surface DLER is available for these

months, but it is a replacement based on other months, which is why the results are not plotted in Fig. 7.

      The four remaining surface type regions ("Amazonian tropical rainforests", "Asian (sub-)tropical forests", "Deciduous forests", and "Grasslands") show a large dependence on viewing angle. For example, for the month May (brown curve) the av-





**Figure 7.** Surface DLER at 772 nm versus viewing angle for nine surface types and four calendar months. The coloured curves represent the average GOME-2 surface DLER. The grey curves provide an indication of the spread in surface DLER over the selected regions.





erage surface DLER in the Amazonian region varies from 0.21 for $\theta_v$ of $-55°$ to 0.36 for $\theta_v$ of $+55°$. For the month November
(blue curve) the increase is from 0.22 to 0.49, which is more than a factor of two. The variability in time is the largest at the
west-looking viewing direction.

   The Asian (sub-)tropical forests and the Deciduous forests on the other hand show a temporal variability which is similar

for the entire viewing angle range. Grasslands show a low temporal variability. For all four vegetated surfaces the anisotropy
of the surface reflection is large. For these surface types the advantage of using DLER instead of LER is therefore substantial.
But also for desert and snow/ice surfaces there is a significant improvement.

## 7   Validation and discussion

In this section the GOME-2 surface DLER database is compared to the established MODIS surface BRDF product. This is

done in two ways. First, in Sect. 7.1, case studies will be performed to analyse the directional behaviour of the two surface
reflectivity products. Then, in Sect. 7.2, global comparisons will be presented. In both sections we make use of the MODIS
MCD43C2 snow-free product, which provides surface BRDF for snow-free land scenes for seven of the MODIS bands. We
select MODIS band 1, centred around 645 nm, as a reference for the 640-nm wavelength band of the GOME-2 surface DLER
database. Based on the results presented in Sect. 3 we may expect only small differences between DLER en BRDF for this

wavelength.

### 7.1   Case studies

In Fig. 8 we present the results from a comparison between GOME-2 surface DLER and MODIS surface BRDF for three
surface type cases. The three reference sites (Amazonian rainforest; equatorial Africa; Libyan desert) were selected primarily
on the basis of their homogeneity. Homogeneity is important because the MODIS MCD43C2 product is provided at a $5\times5$

times higher spatial resolution than the GOME-2 surface DLER database. In all three cases we selected a one-by-one degree
latitude/longitude box, containing 16 grid cells from the GOME-2 surface LER database, and 400 grid cells from the MODIS
MCD43C2 database. The selected grid cells supply all the surface reflectivity parameters that are needed to calculate DLER
and BRDF from the equations that were introduced earlier.

   We then feed the geometry-dependent DLER and BRDF equations with artificial but realistic GOME-2 viewing and solar

angles. The viewing angle $\theta_v$ is varied between $-55°$ and $+55°$ to simulate the scanning motion of the GOME-2 instrument.
This is already sufficient information to calculate the DLER for the 16 selected DLER grid cells (see Eq. (5)). For the BRDF,
the viewing zenith angle is then automatically known ($\theta = |\theta_v|$, see Eq. (4)), but the solar zenith angle $\theta_0$ and relative azimuth
angle $\phi - \phi_0$ also need to be known. The solar angles $\theta_0$ and $\phi_0$ can be determined from solar position calculations (Michalsky,
1988), by specifying an hour angle determined from the GOME-2 equator overpass time of 09:30 LT, taking into account the

change in hour angle because of the displacement in the latitude direction (i.e., caused by the rotation of the Earth) and in
longitude direction (because of the scanning motion from east to west). The viewing azimuth angle $\phi$ is quite a constant factor
(apart from a 180° jump when GOME-2 scans past the exact nadir direction) and was determined from GOME-2 data. With





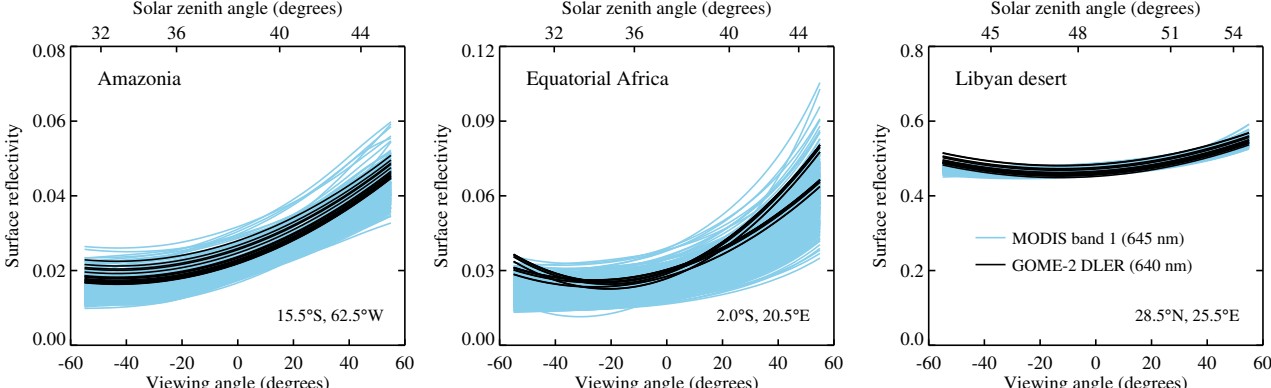

**Figure 8.** Surface reflectivity around 640 nm according to the GOME-2 surface DLER database (black curves) and the MODIS surface BRDF product (blue curves), as a function of the viewing angle, for the Amazonian rainforest (left window), equatorial Africa (middle window), and the Libyan desert (right window). Results are representative for 15 March 2008 and correspond to a one-by-one degree latitude/longitude box with the indicated central coordinates.

all artificial angles known, the MODIS kernels can be calculated and subsequently the surface BRDF can be calculated using Eq. 6 for all 400 MODIS MCD43C2 grid cells.

In the left window of Fig. 8 the scene that is studied is located over the Amazonian rainforest. The blue curves represent the surface BRDF from MODIS band 1 from the 400 MODIS BRDF grid cells. Note that even for what we consider homogeneous

scenes there is already quite some variability between the grid cells. The GOME-2 surface DLER at 640 nm is presented in black. The agreement is good, both qualitatively, in terms of the directional dependence, and in absolute sense. The middle window presents the case of a scene in equatorial Africa. Here the agreement is again good, with the correct dependence on the viewing angle. For the east-viewing directions ($\theta_v < -50°$), however, the agreement seems to be a little less good. It should be noted that there is considerable variability in the 400 blue curves, and that some of the blue curves also show the same upward

bend for $\theta_v < -50°$ as the black curves.

Finally, in the third window of Fig. 8 the scene studied is over the Lybian desert. The Lybian desert is known to be rather stable and is often used as calibration reference site (e.g. Tilstra et al., 2005). The variability in the approximately 100 km by 100 km large box is low. The agreement between DLER and BRDF is good, with small 3–4% differences for the east-viewing directions ($\theta_v < -50°$). The lower performance at the most extreme viewing angles is most likely a result of the fact that in the

15 parabolic fitting procedure explained in Sect. 2.2 inaccuracies are not corrected at the swath ends, but they are in the centre of the swath. Also, the parameterisation used for the DLER is a second-order polynomial, so higher-order dependencies are not described well.

We conclude that the DLER follows the correct directional behaviour. In the next section global comparison are performed, to be able to draw more quantitative conclusions.

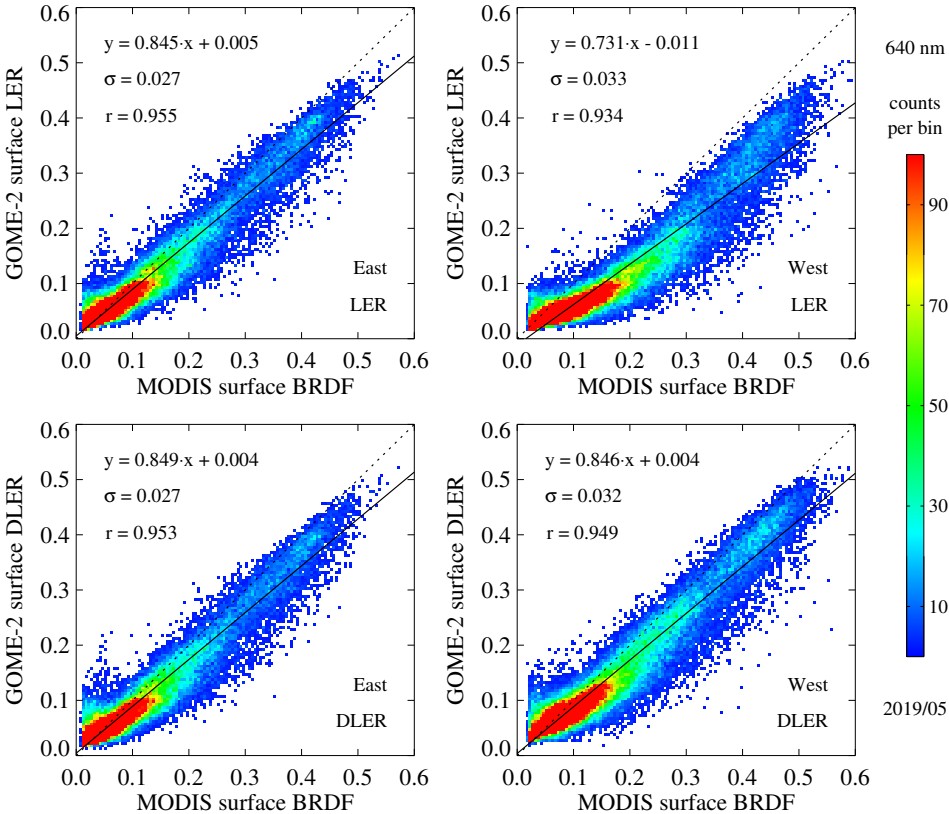

**Figure 9.** Global comparisons between GOME-2 surface (D)LER and MODIS surface BRDF for the period 10–19 May 2019. Top row: GOME-2 non-directional surface LER at 640 nm versus MODIS surface BRDF from band 1 (centred around 645 nm), for eastern and western sides of the orbit swath (see main text). Bottom row: GOME-2 directional surface DLER versus MODIS surface BRDF. For the western side of the orbit swath the directional database agrees much better with MODIS than the non-directional one.

## 7.2 Global comparisons

This section presents results from global comparisons between GOME-2 surface DLER and MODIS surface BRDF. This time, the scattering geometry defined by $\theta$, $\theta_0$, and $\phi - \phi_0$ is prescribed by real GOME-2 observations. For these GOME-2 observations the surface DLER is calculated from the closest grid cell of the GOME-2 surface DLER database, for the appropriate

5    viewing angle $\theta_v$ (see Sect. 7.1). The MODIS surface BRDF is taken from the MODIS MCD43C2 snow-free product and based on the exact 25 grid cells that coincide with the grid cell of the GOME-2 surface DLER database. The MODIS surface BRDF is calculated for the prescribed $\theta$, $\theta_0$, and $\phi - \phi_0$ and averaged over the 25 grid cells. Only observations from the descending forward scan and over land are accepted. Scenes over ocean or scenes containing snow or ice are skipped. Observations with absolute latitudes above 60 degrees are also skipped.


A typical outcome for 640 nm is presented by Fig. 9. The result was obtained for 10 days of GOME-2B observation geometries (10–19 May 2019). In the top row the traditional, non-directional GOME-2 surface LER is presented against the MODIS surface BRDF, for the eastern side of the orbit swath (IndexInScan 1–8; $\theta_\mathrm{v} < -23°$; left window) and the western side of the orbit swath (IndexInScan 17–24; $\theta_\mathrm{v} > +23°$; right window). For the eastern side of the orbit swath the correlation between

surface LER and MODIS BRDF is quite fair. The linear fit has slope 0.845, which admittedly deviates from one, but Pearson's correlation coefficient $r$ is 0.955, indicating good correlation. The standard deviation of the data points w.r.t. the linear fit, $\sigma$, amounts to 0.027. This value of $\sigma$ is in line with uncertainties of ∼0.02 that were reported for the GOME-2 surface LER database and for a few other surface LER databases (Tilstra et al., 2017).

For the western side of the orbit swath, Pearson's $r$ still suggests a reasonable correlation between the two datasets but the

linear fit deviates quite a bit more from the one-to-one relationship. Also, the scatter plot suggests that there is no pure linear relationship between the two databases. The explanation for this behaviour is the bias in the traditional LER databases towards the geometries with the lowest surface LER values. For GOME-2 this affects mostly the western side of the orbit swath, as reported earlier by Lorente et al. (2018).

In the bottom row of Fig. 9 the results are presented for the comparison between the directional GOME-2 surface DLER

and MODIS surface BRDF. For the eastern side of the orbit swath (left window) there are no clear changes compared to the situation in the upper left window. The slope and intercept have improved only marginally. For the western side of the orbit swath (right window), however, the correlation has improved considerably. Both slope and intercept have improved, Pearson's $r$ indicates higher correlation and $\sigma$ went down mildly. More importantly, the eastern and western side of the orbit swath now appear to perform equally well.

The analysis presented in Fig. 9 was repeated for each calendar month for the period 2012–2019 to search for time dependencies in the results. Clear time dependencies, either seasonal or annual, were not found. We also performed the analysis using the SCIAMACHY surface LER database instead of the GOME-2 surface DLER database. The SCIAMACHY surface LER database is by definition a non-directional database, so we could only produce results as in the top row of Fig. 9. The resulting scatter plots were very comparable. For instance, the linear fit to the data had slope 0.860 and intercept 0.005, which

agrees with the numbers in Fig. 9 (0.845 and 0.005, respectively). This may suggest that the deviation of the slope from one is not just specific for GOME-2 but specific for the differences between the LER databases in general and MODIS BRDF.

In the past, small radiometric calibration errors in the GOME-2 level-1 data have been reported (Cai et al., 2012; Wu et al., 2014; Tilstra et al., 2014, 2017). However, these were mainly found for the UV wavelength range, not so much for the particular wavelength that was studied here (640 nm). Also, other studies have reported good agreement with AVHHR and AATSR for

the wavelength range 630–670 nm (Latter et al., 2011). This suggests that the results and conclusions of this section were not significantly influenced by calibration errors in the GOME-2 data.



# 8 Conclusions

In this paper we introduced the directionally dependent Lambertian-equivalent reflectivity (DLER) of the Earth's surface, retrieved from GOME-2 observations. This directional GOME-2 surface DLER database is a major update of the previous non-directional GOME-2 surface LER database in the sense that it describes the anisotropy of surface reflection, while the
traditional LER database considers surface reflection to be isotropic. The DLER database can be used in atmospheric trace gas, aerosol and cloud retrieval algorithms, just like the previous LER database. The retrieval of DLER was described and the anisotropy of the surface reflection could be studied. The anisotropy is especially large for the longer wavelengths and for vegetated surfaces.

Other improvements to the GOME-2 surface DLER database were also described. These include additional wavelength
bands, an improved atmospheric correction taking absorption by oxygen and/or water vapour into account, a higher quality due to the use of more mission data, and a higher spatial resolution. The higher spatial resolution was achieved without compromising the quality by adopting a dynamic gridding approach. However, the main improvement is in the directional nature of the DLER.

To analyse the newly defined property, we conducted a series of radiative transfer simulations to study the theoretical
differences between DLER and BRDF. The study showed, that DLER and BRDF are different for the shorter wavelengths ($\lambda < 500\,\mathrm{nm}$). Here the DLER is meant to be used only in combination with a radiative transfer code that includes Lambertian surface reflection. Lambertian surface reflection is the simplest form of surface reflection and probably the most used in practice. For the longer wavelengths, the differences between DLER and BRDF are small ($500\,\mathrm{nm} < \lambda < 1000\,\mathrm{nm}$) to negligible ($\lambda > 1000\,\mathrm{nm}$) and DLER can effectively be used as a BRDF (and BRDF as DLER).

This conclusion allowed us to compare the GOME-2 surface DLER at $640\,\mathrm{nm}$ with MODIS surface BRDF from MODIS band 1 (centred around $645\,\mathrm{nm}$). A few case studies illustrate that the angular dependencies of DLER and BRDF are indeed comparable, and that there is good agreement between DLER and BRDF. After that, extensive global comparisons confirm that there is indeed good systematic agreement. Moreover, the comparison with MODIS BRDF is also performed using the traditional, non-directional LER database. This LER database performs rather badly at the western side of the GOME-2 orbit
swath. This is in line with findings by Lorente et al. (2018), who found that traditional, non-directional LER databases underestimate the surface reflection for certain viewing geometries. In particular, for the GOME-2 orbit geometry it was found that the surface reflection at $772\,\mathrm{nm}$ was underestimated by a factor of two at the western side of the orbit swath. This is in agreement with our findings. For the western side of the orbit swath, the DLER performs considerably better than the LER database.

The directional DLER database is, in summary, an important improvement on the traditional non-directional LER databases
that are often used in atmospheric retrieval applications. The GOME-2 surface DLER database can be used as input parameter for atmospheric retrieval algorithms working on data from all polar satellites with an equator crossing time close to that of GOME-2. This includes instruments such as GOME, SCIAMACHY, and GOME-2 itself, as well as the future Sentinel-5/UVNS instrument which is scheduled for launch in 2023.





*Data availability.* The GOME-2 surface DLER database can be downloaded from the TEMIS website via the following URL: http://www.temis.nl/surface/albedo/gome2_ler.php.

## Appendix A: Kernels for the Ross–Li BRDF model

This appendix lists the equations needed to calculate the kernels that make up the Ross–Li BRDF model of surface reflectance.

Proper derivations of the Ross–Thick and Li–Sparse kernels can be found in Wanner et al. (1995).

### A1 Ross–Thick volumetric kernel

The Ross–Thick volumetric scattering kernel is defined in the following way (Roujean et al., 1992):

$$K_{\text{vol}} = \frac{(\pi/2 - \xi)\cos\xi + \sin\xi}{\cos\theta + \cos\theta_0} - \frac{\pi}{4} \; . \tag{A1}$$

In Eq. (A1), $\theta$ refers to the viewing zenith angle and $\theta_0$ to the solar zenith angle. The angle $\xi$ is defined according to

$$\cos\xi = \cos\theta\cos\theta_0 + \sin\theta\sin\theta_0\cos(\psi - \psi') \; , \tag{A2}$$

where $\psi$ and $\psi'$ are the viewing and solar azimuth angles following the definition in Strahler et al. (1999). Exact backscattering ($\xi = 0°$) occurs for $\psi - \psi' = 0°$, which agrees with the definition used for the GOME-2 data products.

### A2 Li–Sparse geometric kernel

The Li–Sparse geometric scattering kernel (Li and Strahler, 1986) is defined as:

$$K_{\text{geo}} = O - \sec\theta^\star - \sec\theta_0^\star + \frac{1}{2}(1 + \cos\xi^\star)\sec\theta^\star\sec\theta_0^\star \; . \tag{A3}$$

The term $O$ in Eq. (A3) and the starred angles $\theta^\star$, $\theta_0^\star$, and $\xi^\star$ are calculated using the following set of equations:

$$\theta^\star = \arctan\left(\frac{b}{r}\tan\theta\right) \; , \quad \theta_0^\star = \arctan\left(\frac{b}{r}\tan\theta_0\right) \; , \tag{A4}$$

$$\cos\xi^\star = \cos\theta^\star\cos\theta_0^\star + \sin\theta^\star\sin\theta_0^\star\cos(\psi - \psi') \; , \tag{A5}$$

$$O = \frac{1}{\pi}(t - \sin t\cos t)(\sec\theta^\star + \sec\theta_0^\star) \; , \tag{A6}$$

$$\cos t = \frac{h}{b}\frac{\sqrt{D^2 + (\tan\theta^\star\tan\theta_0^\star\sin(\psi - \psi_0))^2}}{\sec\theta^\star + \sec\theta_0^\star} \; , \tag{A7}$$

$$D = \sqrt{\tan^2\theta^\star + \tan^2\theta_0^\star - 2\tan\theta^\star\tan\theta_0^\star\cos(\psi - \psi')} \; . \tag{A8}$$

The parameters $b/r$ and $h/b$ are the crown relative shape and the crown relative height, respectively. These were fixed to 1 and 2, respectively, following Strahler et al. (1999).



## Appendix B: Dynamic gridding

For the version of the GOME-2 surface LER database described in Tilstra et al. (2017), v1.7, the spatial resolution was $1° \times 1°$. This is a relatively low spatial resolution, leading to artefacts, especially near coastal areas. In Fig. B1 this is illustrated by panels (a) and (b). Panel (a) presents the GOME-1 surface LER at 772 nm for March, for western Europe. Panel (b) presents the v1.7 GOME-2 surface LER. There are differences, mostly because for GOME-2 the MODE-LER field was plotted but for GOME-1 the MIN-LER field. The MODE-LER does a better job at detecting the snow-covered areas. Apart from these differences, both databases, having identical spatial resolutions, show similar difficulties near the coastline. The coastline seems to be pushed land inwards. This is a direct consequence of the way the retrievals operate. By focusing on the smallest scene LER values collected in a grid cell, the observations over water are favoured over those over land.

To remedy the artefacts, we first calculate the surface LER for three spatial resolutions: $1° \times 1°$, $0.5° \times 0.5°$, and $0.25° \times 0.25°$. The $0.25° \times 0.25°$ field has the highest resolution and as such offers the best coastal representation. Unfortunately, the smaller grid cell size also results in less measurements per grid cell and in general leads to a lower quality, mostly due to cloud contamination. In most cases, trading quality for a higher spatial resolution is not desirable. For coastal areas, however, the higher spatial resolution takes precedence.

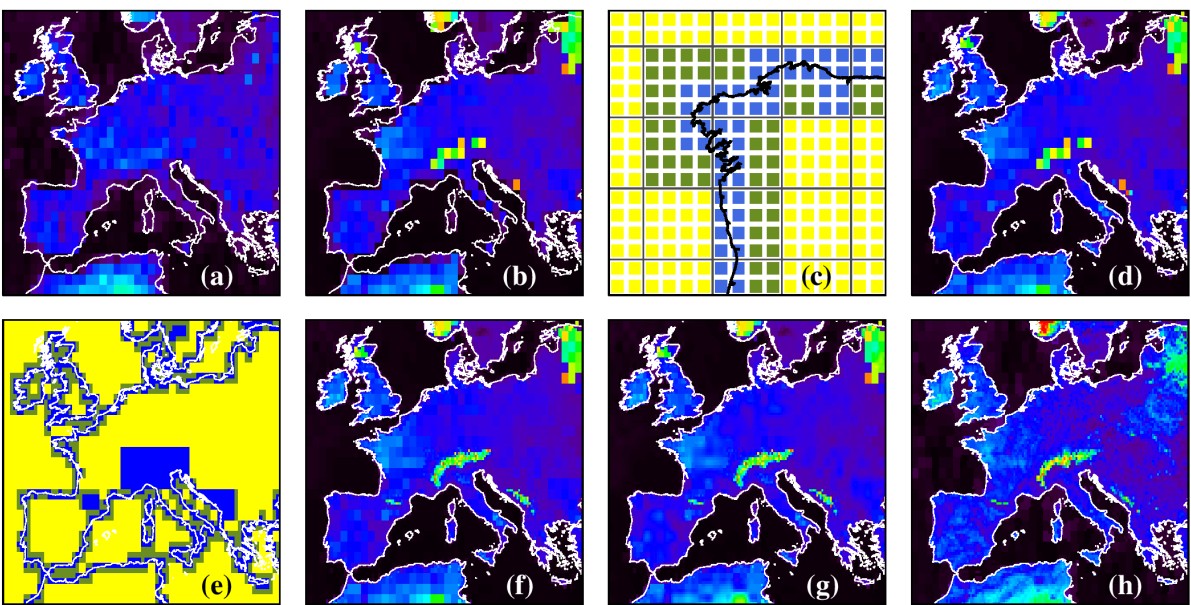

**Figure B1.** Surface LER at 772 nm for the month March from (a) the GOME-1 database, (b) the previous GOME-2 v1.7 database. Panels (c) and (e) help explain the new dynamic gridding procedure described in Appendix B. Panel (d) presents the GOME-2 database with coastline improvement, (f) does the same, but with dynamic gridding also for mountain ranges, (g) does the same, but with the bilinear interpolation scheme applied. Panel (h) presents the MERIS black-sky albedo as a reference for qualitative comparison. For the GOME-1 database the MIN-LER field was plotted, for the GOME-2 database the MODE-LER field.





We start with the $0.25° \times 0.25°$ surface LER field, and use it as a basis. We then perform a loop over all the $0.5° \times 0.5°$ grid cells, and whenever the $0.5° \times 0.5°$ grid cell does *not* contain a coastline, we overwrite the four associated $0.25° \times 0.25°$ grid cells with the surface LER value of the overlapping $0.5° \times 0.5°$ grid cell. Next, we loop over the $1° \times 1°$ grid cells, and whenever the $1° \times 1°$ grid cell does not contain a coastline, we fill the associated sixteen $0.25° \times 0.25°$ grid cells with the

surface LER value of the overlapping $1° \times 1°$ grid cell. The result is a database field that has an intrinsic resolution of $1° \times 1°$ for most of the grid cells, but an intrinsic resolution up to $0.25° \times 0.25°$ near coastlines. This dynamic gridding procedure is illustrated in panel (c) of Fig. B1 for a part of the Portuguese coastline. Blue squares represent $0.25° \times 0.25°$ grid cells, green squares represent $0.5° \times 0.5°$ grid cells, and yellow squares represent $1° \times 1°$ grid cells.

The coastline detection is performed using the Global Self-consistent, Hierarchical, High-resolution Geography (GSHHG)

database (Wessel and Smith, 1996). The GSHHG database offers coastline information for the continents, islands, lakes, rivers, river-lakes, island-in-lakes, and even on the "pond-in-island-in-lake" level. We make use of the highest resolution available of the database, which is the "full resolution" version, available at https://www.soest.hawaii.edu/pwessel/gshhg/. For Antarctica we only consider the grounding coastline as a coastline. We do not take rivers and canals into account as these have negligible surface areas. We do take the so-called river-lakes into account. Islands with an area of less than 5000 km are not taken into

account, nor coastlines from a "pond-in-island-in-lake".

The improvement for coastal areas is demonstrated by panel (d) of Fig. B1. Next, we focus our attention on the snow-covered mountain ranges in panel (d), which are captured poorly because of the low spatial resolution. For these regions, we manually assign rectangular parts of the grid as regions for which the intrinsic resolution should be fixed to $0.25° \times 0.25°$. In other words, in the dynamic gridding procedure these grid cells are protected such that they cannot be overwritten by the contents of the

larger $0.5° \times 0.5°$ and $1° \times 1°$ grid cells. This procedure is illustrated in panel (e) of Fig. B1. The regions containing mountain ranges are indicated in blue. The resulting surface LER field is shown in panel (f) of Fig. B1. The Alps, Pyrenees, and Dinaric Alps are represented much better.

The approach described above leaves us with a database grid of $0.25° \times 0.25°$ resolution that offers a higher resolution near coastlines and for snow-covered mountain ranges. However, for most of the regions over land and ocean the intrinsic resolution

is $1° \times 1°$ and the surface LER grid is filled with redundant information in the form of $4 \times 4$ cell blocks in which 16 identical surface LER values are stored. This is not necessarily wrong, but it does complicate the interpolation that users need to apply to determine the surface albedo for the measurements footprints they are dealing with.

To make things easier for the user the surface LER inside the $4 \times 4$ blocks is distributed over the 16 grid cells using standard bilinear interpolation over the 2D surface LER grid. Care is taken to only perform the interpolation inside and between grid

cells that have an intrinsic resolution of $1° \times 1°$ (i.e., the yellow grid cells in panels (c) and (c) of Fig. B1). Other grid cells, such as the ones near the coastline or mountain ranges, are left untouched. After the bilinear interpolation a common additive correction factor is applied to the 16 grid cells in such a way that the average surface LER of these 16 grid cells is the same as before applying the bilinear interpolation. This step is needed, because we do not want part of the reflectivity of the surface to disappear from the $4 \times 4$ blocks as a result of the bilinear interpolation. Next, we repeat the bilinear interpolation and apply

the resulting additive correction factor to achieve a slightly higher level of smoothness, making the second 2D field a bit more



convincing than the first one. The result is shown in panel (g) of Fig. B1. It is important to stress that the above procedure is not an attempt to artificially increase the spatial resolution. It simplifies the interpolation that needs to be performed by the users.

Finally, in panel (h) of Fig. B1 we present the MERIS black-sky albedo at 775 nm for comparison. The MERIS database has the same resolution as the GOME-2 surface LER database grid, so it can be used well for a qualitative comparison. Coastline

5   and snow-covered mountain ranges compare quite well.

*Author contributions.* LGT wrote the manuscript, developed the algorithms, and performed the validation. ONET performed data processing and supported development. PW calculated the FRESCO cloud product based on the DLER database and analysed the directional behaviour of the cloud properties. PS and PW helped with the radiative transfer modelling. All authors discussed the results and commented on the manuscript.

10   *Competing interests.* The authors declare that they have no conflict of interest.

*Acknowledgements.* The work that was presented in this paper was supported by EUMETSAT via the CDOP-3 project of the AC SAF. EUMETSAT is also acknowledged for providing the GOME-2 data.



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
