# Peer review of "Directionally dependent Lambertian-equivalent reflectivity (DLER) of the Earth's surface measured by the GOME-2 satellite instruments"

_Atmospheric Measurement Techniques, 2020_

## Author Comment (AC1)

**Response to RC1 by Reviewer 3:**

We would like to thank the reviewer for performing a thorough review and for the many helpful suggestions to improve our manuscript.

Below, we respond to each of the review comments. For the sake of clarity, the review comments are given in blue italics and our response is printed in normal font. Changes to the manuscript are printed in green.

*This paper presents a new method to extend the typically used surface LER with a direction dependency. The DLER concept is applied to GOME-2 measurements, and the relation with the surface BRDF is analyzed both theoretically and practically. This paper is well-organized and well-written, and the content is of interest to the community involved in satellite retrieval of atmospheric properties. I recommend publication in AMT after the authors address minor comments below.*

*P1 L21 Please give the full names of the different species.*

We think the chemical notations should be kept because of consistency with the rest of the paper. We propose to give the full name plus chemical notation if the name is not straightforward.

So, we have changed the sentence into:

" This includes the retrieval of trace gases such as ozone ($O_3$), nitrogen dioxide ($NO_2$), bromine oxide (BrO), formaldehyde ($CH_2O$), water vapour ($H_2O$), carbon dioxide ($CO_2$), carbon monoxide (CO), and methane ($CH_4$), and of cloud and aerosol information. "

*P2 Could you also comment on the pros and cons of the Loyola et al., 2020 method with respect to DLER?*

The text has been rewritten and a more extensive discussion of (and comparison with) the GLER and GE_LER databases is now provided in the updated version of the manuscript. Here we also take into account the review comment CC1 by Diego Loyola. The text now reads:

" Recently, several different approaches have been introduced to address this issue. One example is the introduction of geometry-dependent surface Lambertian-equivalent reflectivity (GLER) [Vasilkov et al., 2017; Qin et al., 2019]. In the GLER approach, surface BRDF information from the MODIS surface BRDF database [Gao et al., 2005] is used to calculate Lambertian surface albedo at 466 nm for land-covered satellite footprints of the OMI instrument. The result is a Lambertian surface albedo, ready to be used in a radiative transfer code with Lambertian surface reflection, calculated for the exact scattering geometry of the OMI footprint and for the specific date of the OMI footprint. The advantage is that this Lambertian surface albedo is adjusted to the geometry of the observation, whereas the surface albedo available in the typical Lambertian surface albedo climatologies is more representative for the minimum value of the surface reflectivities that were observed [see e.g. Lorente et al., 2018; Liu et al., 2020] – and therefore underestimates the surface albedo for many of the scattering geometries. The disadvantage of the GLER approach is that it, at least for land-covered scenes, depends fully on the MODIS surface BRDF database. This limits the spectral usage to the seven wavelength bands of the MODIS BRDF product. For the retrieval of $NO_2$ and of cloud properties from the $O_2$-$O_2$ band, both performed in the spectral regime close to 466 nm, this is not a problem – but for many other retrievals it is.

A second example of a geometry-dependent surface LER database is the geometry-dependent effective Lambertian-equivalent reflectivity (GE_LER) database introduced in a recent paper by Loyola et al. [2020]. The GE_LER approach does not depend on external data such as MODIS BRDF and uses machine learning techniques to retrieve the surface reflectivity from level-1 data of the sensor (GOME-2, TROPOMI, or another UVN sensor). Like the GLER, the GE_LER provides daily maps of the surface properties. Unlike the GLER, the GE_LER provides information for all surface types (land, ocean, snow/ice) and covers the UV-VIS-NIR spectral region.

In this paper we introduce the directionally dependent Lambertian-equivalent reflectivity (DLER) of the Earth's surface derived from GOME-2 observations. The surface DLER is retrieved as a function of the viewing geometry and therefore describes the anisotropy of the surface reflectivity. The DLER approach is very different than the GLER approach in that we perform a retrieval directly on GOME-2 level-1 data, not relying on BRDF input (or any other input) from an external database. In this way the wavelength bands, 26 in total, can be chosen freely, allowing the resulting DLER database to support the retrieval of most atmospheric species. A difference compared to the GLER and GE_LER databases is that the directional dependence of the DLER is provided as a parameterisation of the viewing angle. It is not mapped on a satellite footprint and serves as a climatological dependence. The directional approach of the GOME-2 surface DLER is therefore applicable to all polar satellites with equator crossing times close to that of GOME-2 (09:30 LT). This includes satellite instruments like GOME and SCIAMACHY, GOME-2 itself, and the future Sentinel-5/UVNS instrument scheduled for launch in 2023.

Like the GLER and GE_LER, the DLER is a Lambertian property and therefore can be used in situations where radiative transfer calculations include Lambertian surface reflection. . . . "

*P9 L11 The spatial resolution has changed to 40 km \*40 km for GOME-2A. How large is the influence of combining measurements from two instruments with different spatial resolution? Is the instrumental degradation an issue in the DLER retrieval?*

GOME-2A data from 2013 and later years are not taken into account. There would have been quite an impact on the database if these data would have been taken into account. Not so much by the change in spatial resolution, but much more because of the reduction in the orbit swath (from 1920 km to 960 km). The change in orbit swath would have affected the traditional, non-directional LER retrieval mostly, because it is impacted directly by the change in the range of viewing geometries.

Because of the importance of consistency, we have decided not to include GOME-2A data from 2013 and later years. Another reason for not including the data is the lower quality of GOME-2A data from the later years. Also, the GOME-2A orbit has been drifting since June 2017.

Instrument degradation is an important issue that needs to be addressed properly. We correct for instrument degradation in the manner mentioned in [Tilstra et al., 2017] and described more extensively in [Tilstra et al., 2012]. Without a proper degradation correction the retrieved surface LER/DLER would be unusable for at least the shorter (UV) wavelength range.

To clarify the use of GOME-2A data in the database, we added the following sentence to item 4 in section 5 of the manuscript:

" Data from MetOp-A were used only until 2013, because in July 2013 the GOME-2A orbit swath was reduced

from the standard 1920 km to 960 km. The reduction of the viewing angle range would have impacted the non-directional LER since it would then have been biased towards the LER values of the inner part of the orbit swath. "

*P12 L31 What is the reason of applying Eq 9 at 758 nm? P12 L29 concludes that 758 nm can be treated monochromatically.*

Yes, the 758-nm wavelength band could well have been treated monochromatically. We decided to go further for the 758-nm and 772-nm wavelength bands because of their importance to cloud and aerosol retrieval using the $O_2$-A band. A strong argument for doing the additional work is that the absorption is caused by oxygen, not by water vapour. Because the amount of oxygen in the atmosphere is well known, the (very) small adjustment by taking absorption into account is very reliable and worth the additional effort.

To explain this we have added the following sentence to the manuscript:

" For the 758 and 772-nm wavelength bands a monochromatic calculation would have sufficed, but because of their strong importance to cloud and aerosol retrieval using the $O_2$-A band we decided to go further than necessary by adopting spectral calculations. "

*P14 L26 Maybe it is good to show surface albedo maps here?*

We agree and have now added maps of the surface LER to Figure 6. These are shown in Figure 1 of this AC.

**References:**

Gao, F., Schaaf, C. B., Strahler, A. H., Roesch, A., Lucht, W., and Dickinson, R.: MODIS bidirectional reflectance distribution function and albedo Climate Modeling Grid products and the variability of albedo for major global vegetation types, J. Geophys. Res., 110, D01104, doi:10.1029/2004JD005190, 2005.

Liu, S., Valks, P., Pinardi, G., Xu, J., Argyrouli, A., Lutz, R., Tilstra, L. G., Huijnen, V., Hendrick, F., and Van Roozendael, M.: An improved air mass factor calculation for nitrogen dioxide measurements from the Global Ozone Monitoring Experiment-2 (GOME-2), Atmos. Meas. Tech., 13, 755–787, doi:10.5194/amt-13-755-2020, 2020.

Lorente, A., Boersma, K. F., Stammes, P., Tilstra, L. G., Richter, A., Yu, H., Kharbouche, S., and Muller, J.-P.: The importance of surface reflectance anisotropy for cloud and NO2 retrievals from GOME-2 and OMI, Atmos. Meas. Tech., 11, 4509–4529, doi:10.5194/amt-11-4509-2018, 2018.

Loyola, D. G., Xu, J., Heue, K.-P., and Zimmer, W.: Applying FP_ILM to the retrieval of geometry-dependent effective Lambertian equivalent reflectivity (GE_LER) daily maps from UVN satellite measurements, Atmos. Meas. Tech., 13, 985–999, doi:10.5194/amt-13-985-2020, 2020.

Qin, W., Fasnacht, Z., Haffner, D., Vasilkov, A., Joiner, J., Krotkov, N., Fisher, B., and Spurr, R.: A geometry-dependent surface Lambertian-equivalent reflectivity product for UV–Vis retrievals – Part 1: Evaluation over land surfaces using measurements from OMI at 466 nm, Atmos. Meas. Tech., 12, 3997–4017, doi:10.5194/amt-12-3997-2019, 2019.

[Figure]

Figure 1: Left: Global maps of the GOME-2 surface LER, for calendar month March and for 772, 670, and 555 nm. Right: Global maps of the surface anisotropy parameter, defined as the difference between GOME-2 surface DLER at viewing angles of +45° and −45°. The surface anisotropy can be large, especially for vegetated surfaces at wavelengths beyond 700 nm. Over the oceans only non-directional surface LER is provided, as explained in Sect. 2.2.

Tilstra, L. G., de Graaf, M., Aben, I., and Stammes, P.: In-flight degradation correction of SCIAMACHY UV reflectances and Absorbing Aerosol Index, J. Geophys. Res., 117, D06209, doi:10.1029/2011JD016957, 2012.

Tilstra, L. G., Tuinder, O. N. E., Wang, P., and Stammes, P.: Surface reflectivity climatologies from UV to NIR determined from Earth observations by GOME-2 and SCIAMACHY, J. Geophys. Res.-Atmos., 122, 4084–4111, doi:10.1002/2016JD025940, 2017.

Vasilkov, A., Qin, W., Krotkov, N., Lamsal, L., Spurr, R., Haffner, D., Joiner, J., Yang, E.-S., and Marchenko, S.: Accounting for the effects of surface BRDF on satellite cloud and trace-gas retrievals: a new approach based on geometry-dependent Lambertian equivalent reflectivity applied to OMI algorithms, Atmos. Meas. Tech., 10, 333–349, doi:10.5194/amt-10-333-2017, 2017.

---

## Author Comment (AC2)

**Response to RC2 by Dr. Ruediger Lang:**

We would like to thank Dr. Lang for performing a thorough review and for the many helpful suggestions to improve our manuscript.

Below, we respond to each of the review comments. For the sake of clarity, the review comments are given in blue italics and our response is printed in normal font. Changes to the manuscript are printed in green.

*The paper by Tilstra et al, Directionally dependent Lambertian-equivalent reflectivity (DLER) of the Earth's surface measured by the GOME-2 satellite instruments, presents the next evolution of the Lambertian-equivalent reflectivity (LER) surface databases, as derived from high spectral resolution grating spectrometers covering the UV, visible and towards the near and short-wave infrared spectral region. LER surface databases derived from such instruments have the advantage to provide their data at significantly more atmospheric window or well controlled absorption wavelengths, i.e. at higher spectral resolution, than the familiar surface databases derived from band imagers like AVHRR, MODIS, Meteosat, or Sentinel-3. However, up to now, the original LER approach assumed homogenous, non-directional reflection of the surface, which is known to lead to significant biases in particular in the backscatter direction.*

*The directional evolution of the LER surface retrieval approach (DLER), applied to the meanwhile considerable GOME-2 data record of more than 10 years from two Metop platforms at 9:30 LT and over an observation angle range of -55 to 55 degrees, is therefore a very significant improvement to the currently existing (and frequently used) LER databases (Tilstra et al., 2017). The results show that the anisotropy is considerable depending on surface types (in particular for vegetation), as is expected, and that at least for such surface types and at large observation angles the previously used LER databases introduces significant biases.*

*The paper presents a comparison to the principally more accurate bi-directional reflectance distribution function (BRDF) approach, e.g. as applied to MODIS observations, which however is limited in its available spectral resolution. The comparison of synthetic data from radiative transfer calculations shows a good correspondence between the two approaches above 500 nm improving towards longer wavelength. A validation comparing BRDF reflectivity values with DLER values for the MODIS 640 nm band confirms the significant improvements of DLER with respect to LER in the backscatter regime (West-viewing for GOME-2 daylight descending orbits).*

*The scientific results presented here are significant and will be of high interest to users of grating spectrometer data in the UV to near infrared. The paper is well written and I can therefore recommend it for publication in AMT, noting a couple of aspects for the authors to consider.*

*One of the main advantages using DLER (and LER) with respect to imager derived BRDF databases is its higher spectral resolution. While the comparison of BRDF and DLER values derived from synthetic Top-Of-Atmosphere (TOA) data identifies the spectral regime in which both perform similar and where not, the validation results of section 7 provides results only at 640 nm. A tabulated statistics of slope, intercepts and correlation wavelength at other MODIS wavelength (in particular towards the blue) would be very helpful for users to decide where to use or not to use DLER for their applications. In particular, since close to 50% of the provided DLER wavelength are in the <500 nm regime. In this respect, a comparison of DLER performance with respect to the frequently used combination of MODIS BRDF and spectral principle components provided*

*by the ESA ADAM surface reflectance database would be of value for follow on studies.*

The choice for 640 nm (MODIS: band 1 centred at 645 nm) was based on two considerations: (1) we need a MODIS wavelength band that coincides roughly with a DLER wavelength band, and (2) the wavelength should be longer than 500 nm because otherwise BRDF and DLER cannot be compared due to their different nature. The only suitable wavelength band meeting these two criteria is 640 nm (MODIS: band 1).

We have changed the text in the introductory paragraph of section 7 in the following way:

" We select MODIS band 1, centred around 645 nm, as a reference for the 640-nm wavelength band of the GOME-2 surface DLER database. The choice for MODIS band 1 is based on the fact that (i) it is close enough to one of the DLER wavelength bands, and (ii) based on the results presented in Sect. 3 we may expect only small differences between DLER en BRDF for wavelengths longer than 600 nm. "

For deciding whether DLER can be used as a BRDF for wavelengths below 500 nm it would be better to look at the results shown in Figure 4 of section 3.3. Because it would be very hard to isolate the "real" (theoretical) difference from the difference caused by other differences between the two databases.

Nevertheless, we have decided to add results from comparisons with other MODIS bands (at 555 and 469 nm) to the Supplement. These results are also shown here in Figure 1 of this AC.

We have added the following sentence to section 7.2 of the manuscript:

" Results for other wavelength bands can be found in Figs. S6 and S7 in the Supplement. "

Note that these additional figures for the shorter wavelengths were also requested by Reviewer 1 in RC3. Reviewer 1 also requested additional figures similar to Figure 8 in the manuscript.

[Figure]

Figure 1: Pixel-to-pixel comparisons for 555 and 463 nm. These figures are new and presented as Figures S6 and S7 in the Supplement. They are to be compared with Figure 9 of the manuscript.

*Also the paper is only discussing in passing DLER results over persistently snow covered (high mountains and polar regions) regions, and is not discussing ocean surfaces (or/and water bodies in general) at all. Both*

*surface types are either missing or filtered out in BRDF land databases like the ones derived from MODIS, because of considerable uncertainties in the BRDF coefficients for snow surfaces so far, or are generally neglected (like ocean colour variation and potentially associated directional effects apart from glint). Appendix B seems to indicate that DLER (like the previous LER database) also provides values over oceans, although this is never explicitly mentioned or even discussed in the body text of the paper, as it seems. It would surely be very interesting to understand how well DLER performs for these two surface types, which are (or seem) both included in the discussed database.*

The GOME-2 surface DLER database does in fact provide values over the oceans, but the polynomial coefficients $c_0$, $c_1$, and $c_2$ are equal to zero over the oceans. This is mentioned in section 2.2. Over the oceans, the database therefore provides the standard non-directional minimum LER value which was already discussed and analysed in our previous paper [Tilstra et al., 2017]. This non-directional value corresponds more to the diffuse (non-specular) component of the surface reflection over water (section 2.2, last sentence).

For snow/ice surfaces the GOME-2 surface DLER database does provide full directional LER values, so the surface anisotropy is contained in the database. Results for snow/ice surfaces are shown in Figures 6 and 7. In Figure 6 the surface anisotropy over snow/ice surfaces is shown by the blue colour (as opposed to the red colour seen for desert/vegetation). In Figure 7 results are shown for Antarctica and Greenland.

Note that we have updated Figure 6 of the manuscript, in response to a review comment made by Reviewer 3 in RC1. Figure 6 now not only shows the surface anisotropy parameter, but also presents global maps of the (non-directional) surface LER. This now makes it more clear that surface LER is also provided for the ocean. We have also updated the caption of Figure 6, which now mentions explicitly that non-directional surface LER is provided over the oceans. The updated figure and caption are shown in Figure 2 of this AC.

*Finally, it is not very clear to me why in the "Case studies" part of the validation section (Section 7.1) the authors emphasize the need for focussing on largely homogenous surfaces. A proper averaging of MODIS BRDF sub-pixels to the DLER grid pixel should in principle provide an accurate comparison independent of sub-pixel surface variations. And it would be also interesting to provide the corresponding averaged MODIS BRDF results in Figure 8 for comparison with the DLER grid pixel results along with the individual ones (and ideally show similar comparisons for non-homogeneous cases too).*

A higher inhomogeneity in the scenes would make the comparisons shown in section 7.1 less significant. For validation it is better to have homogeneous scenes to reduce the spread. For example, the collocation differences between GOME-2 and MODIS can cause spread. It is true that in section 7.2 we perform a proper averaging of the MODIS surface BRDF grid cells to the surface DLER grid cells. However, in section 7.1 we consider pre-defined regions (of one by one degree) and present the viewing angle dependence of the MODIS BRDF and DLER grid cells by plotting these for each of the grid cells. We on purpose do not perform any averaging in section 7.1, so that the images provide information about the variability of the surface within the one-by-one degree region.

The plots show that the inhomogeneity can be relatively large, even for scenes which we consider to be homogeneous. The curves indicate that there is considerable spread due to the inhomogeneity of the surface. This is seen in both MODIS surface BRDF and GOME-2 surface DLER.

[Figure]

Figure 2: Left: Global maps of the GOME-2 surface LER, for calendar month March and for 772, 670, and 555 nm. Right: Global maps of the surface anisotropy parameter, defined as the difference between GOME-2 surface DLER at viewing angles of +45° and −45°. The surface anisotropy can be large, especially for vegetated surfaces at wavelengths beyond 700 nm. Over the oceans only non-directional surface LER is provided, as explained in Sect. 2.2.

To be able to draw strong conclusions from comparing MODIS BRDF and GOME-2 DLER, the spread should be as small as possible. Nevertheless, the results in section 7.1 are to be considered qualitative results, where the results from section 7.2 are to be considered quantitative results.

**References:**

Tilstra, L. G., Tuinder, O. N. E., Wang, P., and Stammes, P.: Surface reflectivity climatologies from UV to NIR determined from Earth observations by GOME-2 and SCIAMACHY, J. Geophys. Res.-Atmos., 122, 4084–4111, doi:10.1002/2016JD025940, 2017.

---

## Author Comment (AC4)

**Response to RC3 by Reviewer 1:**

We would like to thank the reviewer for performing a thorough review and for the many helpful suggestions to improve our manuscript.

Below, we respond to each of the review comments. For the sake of clarity, the review comments are given in blue italics and our response is printed in normal font. Changes to the manuscript are printed in green.

This work is a great addition to the currently available angular dependent LER products that have considered surface BRDF effects based on the LER concept. It provides important improvements to the author's previous LER climatology product based on GOME-2 observations, which include directional LER (DLER) at as many as 26 wavelengths from 328 to 772 nm. It fits well to the scope of AMT and should be considered publishing after addressing the following issues.

1. Although bidirectional reflectance distribution function (BRDF) mathematically describes the scattering of a parallel beam of incident light from one direction in the hemisphere into another direction in the hemisphere, the BRDF itself, as a ratio of infinitesimals, is a derivative with "instantaneous" [in angle] values that can never be measured directly, as stated in Nicodemus et al. (1977).

Therefore, it seems more proper to use BRF or simply reflectance in many places throughout the manuscript when referring to directional reflectance datasets. BRF (Bidirectional Reflectance Factor) is defined as the ratio of the radiant flux reflected by a surface to that reflected into the same reflected-beam geometry by an ideal (lossless) and diffuse (Lambertian) standard surface, irradiated under the same conditions.

So in remote sensing community, BRF is commonly used to describe the reflectance factor calculated from either ground measurements or satellite observations that have finite field-of-view (FOV). The authors can refer to Schaepman-Strub et al., (2006) for more details regarding use of BRDF and BRF.

The suggestion here is to only keep BRDF before 'model', 'parameter', 'product', 'database' etc (i.e., only use it as an adjective) and replace BRDF with BRF in other places when referring to directional data set itself, such as at line 9, 11, 14 in page 1, as well as many instances in the rest of the manuscript.

The reviewer is right in principle, but for the clarity of the paper we choose to use only BRDF and not BRF as well. We have, however, checked the manuscript for the use of the term "BRDF" and now use it as an adjective instead or have replaced it with "reflectance" for a number of cases where this is more appropriate. For instance, the lines 9–14 on page 1 of the manuscript now read:

"The relation between DLER and BRDF surface reflectance is studied using radiative transfer simulations. For the shorter wavelengths ( $\lambda < 500$  nm), there are significant differences between the two. For instance, at 463 nm the difference can go up to 6% at 30° solar zenith angle. The study also shows that, although DLER and BRDF surface reflectances are different properties, they are comparable for the longer wavelengths ( $\lambda > 500$  nm). Based on this outcome, the GOME-2 surface DLER is compared with MODIS surface BRDF data from MODIS band 1 (centred around 645 nm), using both case studies and global comparisons. The conclusion of this validation is that the GOME-2 DLER compares well to MODIS BRDF data ... "

In section 1 and also in section 2.2 we now also refer to the paper by Schaepman-Strub et al. [2006].

2. It seems not proper by calling LER (defined in Eq.2) as albedo (see line 21, page 3) because albedo is defined as the ratio of the radiant flux reflected into the whole upper hemisphere (i.e., the integal from all viewing angles) to the incident radiant flux. In other words, albedo is independent of viewing directions, but LER does for a non-Lambertian surface. That is the physical basis for DLER in this study, GE\_LER (Loyola et al., 2020) and GLER (Vasilkov et al., 2017). Also as shown in Eq.3, LER has the same unit as R.

We agree that LER should not be called an albedo, however, in the derivation leading up to equation (3), the term "albedo" is only used for the surface albedo, which is explicitly defined to be independent on the viewing directions. In equation (2), the LER is not defined yet.

We have changed the last sentences of section 2.1 in the following way:

"Both parameters  $R_{\lambda}^{\text{obs}}$  and  $R_{\lambda}^{0}$  depend on  $\mu$ ,  $\mu_{0}$ , and  $\phi - \phi_{0}$ , so, in general, so does  $A_{\text{s}}$ . When clear-sky conditions apply, the parameter  $A_{\text{s}}$  is the Lambertian-equivalent reflectivity (LER) of the surface."

This avoids referring to the LER as an albedo. Additionally, we have searched the entire manuscript for the words "surface albedo" and have replaced these with "surface reflectance" where appropriate.

3. The authors should mention in the figure caption that Fig.1b only applies to land surfaces because reflection from a non-Lambertian surface, in general, could have another peak in the forward scattering direction, i.e., specular (mirror) reflection over snow/ice or water surfaces (so-called sunglint) in addition to the hot spot peak (retroreflection) over rough surfaces like vegetation due to shadow hiding.

The caption mentions that the surface reflection distribution in Figure 1b is representative for vegetation. To avoid confusion, we changed the caption of Figure 1. The caption now reads:

"Left: Illustration of the principle of Lambertian (isotropic) surface reflection. Right: Surface reflection distribution with a retroreflection lobe, representative for land surfaces covered by vegetation. In the DLER retrieval code, the orbit swath is divided into five viewing angle ranges and for each segment the surface LER is determined in the usual way."

4. The authors should provide reference(s) to justify the use of a parabolic function to simulate directionality of surface reflection for vegetated surfaces as shown in Eq.5. There have been many publications from BRDF modeling community in land remote sesing that demonstrated such parabolic function only applies to non-vegetated surfaces such as desert or bare soil. That's why in MODIS BRDF products, a linear combination of different kernels is used to describle surface drectional reflection in general as described by Eq.6.

A second-order polynomial can indeed not catch all characteristics for all geometries and for all surface types. In particular, the "hot spot" will not be represented well for all geometries. However, compared to the standard non-directional approach the use of second-order polynomial is already quite an improvement.

We have experimented with a higher number of bins, to get a better feel of the variability of the DLER. The higher number of bins goes at the expense of the reliability of the LER in the bins, because of a higher probability that a good cloud-free value can be found. For some regions, however, this does work and we can analyse the variability of the DLER.

We did indeed see that the parabolic shape works very well for desert surfaces. For vegetation the parabolic

function seemed to work well as well. Note that the regime of the viewing and solar angles that are involved is representative for the GOME-2 orbit, and therefore constitutes a subset of the range of angles supported by MODIS BRDF. Also, the GOME-2 orbit swath is less wide than that of the MODIS instruments. It may be so that the geometries involved for GOME-2 are more forgiving when it comes to modelling the directionality of the surface reflection with a simple parabolic function as we did.

In section 5, item 5, we now mention more clearly that the DLER is an approximation:

" 5. The database now offers directionally dependent surface LER (DLER). This means that the anisotropy of the surface reflection, often called the BRDF effect, is contained in (and described by) the DLER database. The provided DLER is an approximation in the sense that the second-order polynomial approach presented in section 2.2 in combination with the 5 angular bins of about 20 degrees each will not be able to catch the angular variability of the DLER for all surface types and situations. In particular, for vegetated surfaces the "hot spot" will not in all circumstances and geometries be represented well. Also, the DLER database is in principle representative only for the geometry of the GOME-2 orbit. "

**5. The title of section 3.1 should be changed to 'MODIS BRDF model'.**

Agreed, we have changed the title of section 3.1 to "MODIS BRDF model":

" 3.1 MODIS BRDF model "

6. Though I believe DLER is derived from the real GOME-2 measurements as mentioned in the 2nd paragraph (lines 9-14, page 4) of section 2.2, it also says DLER product is based on simulated TOA reflectances with DAK (line 23, page 6), That would create some confusion for readers and should be clarified.

DLER is indeed derived from the real GOME-2 measurements. However, in section 3.2, the DLER database is not described. Here the model calculations of DLER and MODIS BRDF are described.

To avoid confusion, we have changed the text in the following way:

"Next, the simulated surface DLER is calculated from the simulated TOA reflectances using a similar setup as the one described in Sect. 2. "

7. Since real GOME-2 observations are used to derive DLER, it is not clear how the aerosol and cloud contaminated data is removed and the data screening criteria used. How data gaps (due to aerosol and/or cloud contaminations) are handled? All these need to be addressed in the manuscript.

Absorbing aerosol is removed by using the Absorbing Aerosol Index (AAI) to filter out absorbing aerosol. However, we do not actively apply a correction for background scattering aerosols, and these are in principle included in the DLER database, even though the algorithm is focused on the minimum scene reflectivities. Cloud screening is performed in a statistical manner, without the use of external cloud information, as described in section 5 of the manuscript (lines 6–10). Post-processing steps are needed to address gaps and residual cloud contamination. All of this is described extensively in our previous paper [Tilstra et al., 2017].

We now mention these steps briefly in the updated version of the manuscript, by referring to our previous paper. The text in section 5 has been changed in the following way: "... the observed scene LER values of a specific month (but from all available years) are distributed onto a latitude/longitude grid which represents the Earth's surface for that specific calendar month. In this step, observations containing absorbing aerosols are filtered out using the Absorbing Aerosol Index (AAI). For each grid cell the distribution of scene LER values is then analysed statistically to find the cloud-free observations. This is done in two ways ...

... After these steps, post-processing corrections are performed that take of issues such as gaps and residual cloud contamination. The above steps and procedures have all been described extensively in Tilstra et al. [2017]. There are ... "

8. It looks like DLER product only has five viewing direction bins since the data is sorted out through five sub-containers (see line 12, page 4). What are the exact angular widths for these five bins, any justification that these five anglular bins can adquently describe the angular distribution of GOME-2 measurements? What is the angular resolution in the GOME-2 x-track scan positions and is that the basis for selecting the five view angle bins? All these need to be deliberated a little more.

Since this is not a full consideration of BRDF effects but a rough approximation, it should be said so in item 5 of page 10 (lines 22-23).

Although this approximation may work for not too large SZAs where the angular width of the hot spot is pretty broad. However, when SZA is high (e.g., larger than 45 deg), the hot spot width becomes very narrow. In such situation if GOME-2 viewing angle falls into the hot spot region, the peak will be smoothed out by the large angular bin as shown in figure 1b, resulting in much smaller DLER value. Same is true for the sunglint effect over ocean should this five viewing angle scheme be applied to water surfaces,

The GOME-2 instrument has a swath width of 1920 km. In the normal operation of the instrument this swath is made up of 24 pixels from east to west (in the forward scan of the instrument). The viewing zenith angle goes up to 55 degrees at the swath edges. The size of the 5 angular bins is about 20 degrees in viewing angle.

The decision to use only 5 bins (and not more) was based on the fact that the algorithm needs enough observations to be able to find the necessary cloud-free observations. If we decide to use more bins, then there will be less data contained in each bin, decreasing the chances of finding cloud-free observations.

It is certainly true that the DLER database provides an approximation of the directionality. This is now mentioned explicitly in the manuscript, in item number 5 of section 5, as suggested:

" 5. The database now offers directionally dependent surface LER (DLER). This means that the anisotropy of the surface reflection, often called the BRDF effect, is contained in (and described by) the DLER database. The provided DLER is an approximation in the sense that the second-order polynomial approach presented in section 2.2 in combination with the 5 angular bins of about 20 degrees each will not be able to catch the angular variability of the DLER for all surface types and situations. In particular, for vegetated surfaces the "hot spot" will not in all circumstances and geometries be represented well. Also, the DLER database is in principle representative only for the geometry of the GOME-2 orbit. "

As for the sun glint effect over the oceans, the climatology does not try to describe the directionality of the surface reflectivity over water surfaces. Over the oceans the directionality is not recorded. That is, the polynomial coefficients  $c_0$ ,  $c_1$ , and  $c_2$  are zero over the oceans. This is explained in section 2.2 of the manuscript.

9. Fig.7 shows results at 772 nm. Readers may also be interesed to see results for short wavelengths (e.g., 363, 380, 340 or shorter) since some of the short wavelengths in Table 1 are widely used in trace gas retrieval based on UV/VIS data such as ozone, NO2, aerosol via aersol index and clouds via O2-O2 or rotational Raman scattering algorithm.

We had selected 772 nm because for this wavelength the largest anisotropy is to be expected. Another reason for selecting 772-nm wavelength band is because it is relevant for cloud and aerosol retrieval using the  $O_2$ -A band. In Figure 1 of this AC we present, as an example, the result for 463 nm.

Figure 1: Similar to Figure 7 of the manuscript, but now for 463 nm.

We have decided to add results for other wavelength bands as part of the Supplement. The wavelength bands that were added to the Supplement are: 670, 463, and 380 nm.

10. For Figs. 8-9, DLER from 645 nm is compared with MODIS band 1. Can the authors show comparisons with the shortest wavelength in MCD43C2 product like band 3 (centered at 470 nm)? Though the difference

would be larger as expected, readers may be interested to see how DLER and MODIS BRF follow with each other in angular distributions.

We have compared DLER to the MODIS BRDF product for three wavelengths:

- 640 nm MODIS band 1 (around 645 nm)
- 555 nm MODIS band 4 (around 555 nm)
- 463 nm MODIS band 3 (around 469 nm)

The results are shown in Figures 2 and 3 of this AC.

Looking at the plots in Figure 2 of this AC, there seems to be an offset for the shortest wavelength. A possible explanation could be the background aerosol scattering, which is still present to some degree in the GOME-2 surface DLER database because there is no active filtering for scattering aerosol in the retrieval code.

---

## Author Response (AR2)

**Author response**

The reviewer is absolutely right and we have resolved the issue in the updated version of the manuscript.

We have changed the text in the Introduction (section 1) in the following way:

" Recently, several different approaches have been introduced to address this issue. One example is the introduction of geometry-dependent surface Lambertian-equivalent reflectivity (GLER) (Vasilkov et al., 2017; Qin et al., 2019; Fasnacht et al., 2019). In the GLER approach, surface BRDF information from the MODIS surface BRDF database (Gao et al., 2005) is used to calculate Lambertian surface albedo at 466 nm for land-covered satellite footprints of the OMI instrument. For the footprints over water surfaces model calculations are used (Fasnacht et al., 2019). The result is a Lambertian surface albedo, ready to be used in a radiative transfer code with Lambertian surface reflection, calculated for the exact scattering geometry of the OMI footprint and for the specific date of the OMI footprint. The advantage is that this Lambertian surface albedo is adjusted to the geometry of the observation, whereas the surface albedo available in the typical Lambertian surface albedo climatologies is more representative for the minimum value of the surface reflectivities that were observed (see e.g. Lorente et al., 2018; Liu et al., 2020) – and therefore underestimates the surface albedo for many of the scattering geometries. The disadvantage of the GLER approach is that it, at least for land-covered scenes, depends fully on the MODIS surface BRDF database. This limits the spectral usage to the seven wavelength bands of the MODIS BRDF product for land-covered scenes. For the retrieval of NO2 and of cloud properties from the O2-O2 band, both performed in the spectral regime close to 466 nm, this is not a problem – but for many other retrievals it is.

A second example of a geometry-dependent surface LER database is the geometry-dependent effective Lambertian-equivalent reflectivity (GE_LER) database introduced in a recent paper by Loyola et al. (2020). The GE_LER approach does not depend on external data such as MODIS BRDF data and uses machine learning techniques to retrieve the surface reflectivity from level-1 data of the sensor (GOME-2, TROPOMI, or another UVN sensor). Like the GLER, the GE_LER provides daily maps of the surface properties. The GE_LER provides information for all surface types (land, ocean, snow/ice) in one database and covers the UV-VIS-NIR spectral region. "

This solves the issue in a number of ways: this section now explicitly mentions that GLER is provided also over water surfaces, it refers to the paper (by Fasnacht et al., 2019) presenting the approach for water surfaces, and the last sentence that (wrongly) claimed that GLER is not provided over the ocean has been changed to only describe the GE_LER.